# Neural population dynamics underlying evidence accumulation in multiple rat brain regions

**Brian DePasquale[1]\*‡, Carlos D Brody[1,2]\*†, Jonathan W Pillow[1,3]\*†**

[1]Princeton Neuroscience Institute, Princeton University, Princeton, United States;
[2]Howard Hughes Medical Institute, Princeton University, Princeton, United States;
[3]Department of Psychology, Princeton University, Princeton, United States

**Abstract** Accumulating evidence to make decisions is a core cognitive function. Previous studies have tended to estimate accumulation using either neural or behavioral data alone. Here, we develop a unified framework for modeling stimulus-driven behavior and multi-neuron activity simultaneously. We applied our method to choices and neural recordings from three rat brain regions—the posterior parietal cortex (PPC), the frontal orienting fields (FOF), and the anterior-dorsal striatum (ADS)—while subjects performed a pulse-based accumulation task. Each region was best described by a distinct accumulation model, which all differed from the model that best described the animal's choices. FOF activity was consistent with an accumulator where early evidence was favored while the ADS reflected near perfect accumulation. Neural responses within an accumulation framework unveiled a distinct association between each brain region and choice. Choices were better predicted from all regions using a comprehensive, accumulation-based framework and different brain regions were found to differentially reflect choice-related accumulation signals: FOF and ADS both reflected choice but ADS showed more instances of decision vacillation. Previous studies relating neural data to behaviorally inferred accumulation dynamics have implicitly assumed that individual brain regions reflect the whole-animal level accumulator. Our results suggest that different brain regions represent accumulated evidence in dramatically different ways and that accumulation at the whole-animal level may be constructed from a variety of neural-level accumulators.

**\*For correspondence:**
bddepasq@bu.edu (BDeP);
brody@princeton.edu (CDB);
pillow@princeton.edu (JWP)

†These authors contributed equally to this work

**Present address:** ‡Department of Biomedical Engineering, Boston University, Boston, United States

## Editor's evaluation

This valuable paper presents findings showing that different brain regions were best described by a distinct accumulation model, which all differed from the model that best described the rat's choices. These findings are solid because the authors present a very strong methodological approach. This work will be of interest to a wide neuroscientific audience.

## Introduction

Accumulation of evidence is a critical process underlying decision-making in complex environments where relevant information is distributed across time. Choice data from evidence accumulation tasks e.g., *Brunton et al., 2013*; *Raposo et al., 2012*; *Sanders and Kepecs, 2012* have allowed for the development of sophisticated models of animals' accumulation strategies (e.g., *Bogacz et al., 2006*; *Brunton et al., 2013*; *Genkin et al., 2021*; *Gold and Shadlen, 2007*; *Ratcliff et al., 2016*; *Ratcliff and McKoon, 2008*; *Shinn et al., 2020*; *Wiecki et al., 2013*). In parallel, neural correlates of accumulated evidence have been found in a wide variety of brain regions (e.g., *Brody and Hanks, 2016*; *Churchland et al., 2011*; *Ding and Gold, 2010*; *Erlich et al., 2011*; *Gold and Shadlen, 2007*; *Hanks et al.,*

*2015*; *Huk and Shadlen, 2005*; *Kim and Shadlen, 1999*; *Mante et al., 2013*; *Ratcliff et al., 2003*; *Roitman and Shadlen, 2002*; *Shadlen and Newsome, 2001*; *Yartsev et al., 2018*) and methods have been developed to describe the statistical relationship between neural activity and accumulated evidence (e.g., *Aoi et al., 2020*; *Beck et al., 2008*; *Churchland et al., 2011*; *Hanks et al., 2015*; *Latimer et al., 2015*; *Latimer and Freedman, 2021*; *Park et al., 2014*; *Zoltowski et al., 2019*; *Zoltowski et al., 2020*).

Obtaining a comprehensive account of how stimulus-influenced accumulated evidence underlies neural activity and subject choice remains an open problem. For example, few analysis methods which use precise spike timing information take into account the timing of stimulus information or use choice data directly (e.g., *Latimer et al., 2015*). Likewise few methods that use the precise timing of stimulus information to infer accumulated evidence use neural responses directly (e.g., *Hanks et al., 2015*). To address this gap, we developed a framework for inferring probabilistic evidence accumulation models jointly from choice data, neural activity, and precisely controlled stimuli.

A complete understanding of decision-making necessitates models that can comprehensively combine stimuli, neural activity, and behavior. The evidence accumulation process inferred from behavioral data alone need not correspond to the accumulation process that best matches data from a single brain region; behavior is the result of interactions between multiple brain regions. For example, two brain regions, one favoring accumulation of early evidence (e.g., an unstable accumulator) and the other favoring accumulation of late evidence (e.g., a leaky accumulator) could together support stable behavior-level accumulation. By fitting accumulator models to neural data from multiple brain regions and to subject choice data, we gained the opportunity to probe for the first time whether different brain regions reflect the same, or different, accumulation processes and how those individual processes correspond to the animal's overall behavior.

We applied our model to choices and neural responses from three brain regions known to be involved in evidence accumulation while animals perform a pulse-based evidence accumulation task. A single variable representing accumulated evidence, shared across neurons within a brain region, accurately accounted for both neural and choice data. We identified distinct signatures of accumulation reflected in each brain region, all of which differed from the accumulation model that best described behavior, supporting the idea that whole-organism accumulation likely results from multiple accumulation processes. Prior analysis of these data found that the anterodorsal striatum (ADS) represented accumulated evidence in a graded manner (*Yartsev et al., 2018*) while the frontal orienting fields (FOF) represented choice more categorically (*Hanks et al., 2015*). Our analysis confirms the ADS as a veracious representation of accumulated evidence while offering a more nuanced view of the FOF: the accumulation model that best described FOF activity was dynamically unstable, producing neural responses that looked like a categorical representation of choice but that were in fact unstable accumulators sensitive to early stimulus information. Additionally, we analyzed recordings from the posterior parietal cortex (PPC), a brain region long studied in connection to evidence accumulation (*Hanks et al., 2015*; *Roitman and Shadlen, 2002*; *Shadlen and Newsome, 2001*), where we identified neural correlates of graded evidence accumulation, albeit more weakly than in the ADS.

Incorporating neural activity into accumulation models reduced the uncertainty in the moment-by-moment value of accumulated evidence when compared to models fit only to animal choices. This reduction in uncertainty led to a more refined picture of the moment-by-moment value of accumulated evidence, which made the model more informative about what choice the animal intended to make. Our model allowed us to implement a novel analysis to examine how subject provisional choice changed during individual trials, commonly referred to as 'changes of mind' (*Boyd-Meredith et al., 2022*; *Kiani et al., 2014*; *Peixoto et al., 2021*), that revealed extensive choice vacillation reflected in ADS activity and greater choice certainty reflected in FOF activity.

Broadly, our framework offers a unified, mechanistic, and probabilistic description of the moment-by-moment accumulation process that underlies decision-making. Our flexible framework offers a computationally efficient method for identifying a key normative decision-making model using multiple types of data, and can easily accommodate simultaneous recordings from many neurons or recordings performed sequentially over many days. It provides a platform for quantitatively characterizing choice-related information in neural responses and can be used to understand how different brain regions implement an accumulation strategy.

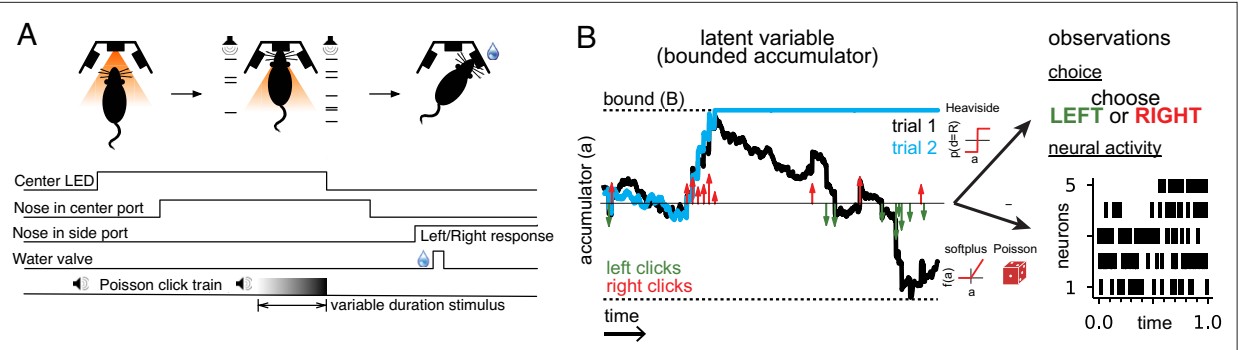

**Figure 1.** Accumulating evidence task and latent variable model. (**A**) Rats performed a pulse-based evidence accumulation task. A central LED illuminates indicating that the rat can begin a trial by poking its nose in a central port. After a delay of random duration, an auditory stimulus of variable duration is delivered—a series of brief auditory pulses played from a left and a right speaker. Upon cessation of the stimulus, the rat must orient to the direction of the greater number of pulses to receive a water reward. (**B**) The model relates the click-based sensory stimulus to two types of observations—the animal's choice and neural activity observed during the task. The latent variable model is a bounded accumulator. Left and right clicks (green and red arrows, respectively) push the variable to one side or the other; if the accumulator variable reaches the bound $B$ (dotted line) accumulation ceases. Seven parameters govern the dynamics of $a(t)$ (see main text). Two different hypothetical trajectories of $a(t)$ are illustrated (black and blue) for the same click stimulus; the two trajectories differ due to the diffusive and stimulus noise in the model. $a(t)$ relates to the animal's choice by a Heaviside step function and to neural activity by way of a softplus nonlinearity and a Poisson distribution. $a(t)$ is common for all simultaneously recorded neurons and each neuron has its own parameters that determine its tuning curve.

The online version of this article includes the following figure supplement(s) for figure 1:

**Figure supplement 1.** Recovering the parameters of synthetic data.

**Figure supplement 2.** Recovering the parameters of synthetic data for multiple datasets.

## Results

We analyzed behavioral and neural data from rats trained to perform a perceptual decision-making task (*Brunton et al., 2013*). Rats listened to two simultaneous series of randomly timed auditory clicks, one from a speaker on the left and one from a speaker on the right. After the end of the click train, the rat had to orient to the side with a greater number of clicks to receive a reward (*Figure 1A*).

We analyzed behavioral choice data and electrophysiological neural recordings from 11 rats. In total, we analyzed 37,179 behavioral choices and 141 neurons from three brain areas—the PPC, the FOF, and the ADS. Prior electrophysiological and lesions studies have shown that these brain regions play a key role in evidence accumulation (*Ding and Gold, 2013*; *Ding and Gold, 2010*; *Erlich et al., 2015*; *Erlich et al., 2011*; *Gold and Shadlen, 2007*; *Gold and Shadlen, 2000*; *Hanks et al., 2015*; *Huk and Shadlen, 2005*; *Kim and Shadlen, 1999*; *Mante et al., 2013*; *Roitman and Shadlen, 2002*; *Shadlen and Newsome, 2001*; *Yartsev et al., 2018*).

Data were collected after the animals were well trained and exhibiting a high level of performance (*Brunton et al., 2013*; *Hanks et al., 2015*; *Yartsev et al., 2018*); these data were collected as part of two earlier studies and have been previously analyzed (*Hanks et al., 2015*; *Yartsev et al., 2018*). Data were subject to a selection criterion for inclusion in our study. We selected neurons with significant tuning for choice during the stimulus period (two-sample t-test, p<0.01) because choice tuning is a prerequisite for reflecting accumulation-like signals. Information about the data is summarized in *Table 1*. Once tuning significance was determined, our dataset consisted of 68 neurons from FOF, with 7382 behavioral choices recorded from five rats over 46 behavioral sessions; 25 neurons from PPC, with 9037 behavioral choices from three rats over 24 sessions; and 48 neurons from ADS, with 10,760 behavioral choices from three rats over 27 behavioral sessions.

### A latent variable model of behavioral choice and neural activity

One of the most common normative models of the internal mental processes that underlie evidence accumulation is the drift-diffusion to bound model (DDM; *Figure 1B*; *Bogacz et al., 2006*; *Brunton et al., 2013*; *Gold and Shadlen, 2007*; *Ratcliff and McKoon, 2008*). While previous work has tended to fit this model (either explicitly or implicitly) using either choice data (e.g., *Brunton et al., 2013*; *Chandrasekaran and Hawkins, 2019*; *Gold and Shadlen, 2007*; *Ratcliff et al., 2016*; *Shinn et al.,*

**Table 1.** Number of neurons, sessions, and trials for each rat.

| Rat | Region | Sessions | Neurons | Trials | Sessions with greater than 1 neuron | Max. # of simultaneously recorded neurons |
|---|---|---|---|---|---|---|
| B068 | FOF | 11 | 13 | 5859 | 2 | 2 |
| T034 | FOF | 9 | 10 | 4138 | 1 | 2 |
| T036 | FOF | 8 | 12 | 3026 | 4 | 2 |
| T063 | FOF | 17 | 32 | 4002 | 9 | 3 |
| T030 | FOF | 1 | 1 | 357 | 0 | 1 |
| T035 | PPC | 15 | 16 | 5919 | 1 | 2 |
| T011 | PPC | 7 | 7 | 2235 | 0 | 1 |
| B053 | PPC | 2 | 2 | 883 | 0 | 1 |
| T080 | ADS | 5 | 6 | 1731 | 1 | 2 |
| T103 | ADS | 19 | 38 | 8332 | 9 | 5 |
| E021 | ADS | 3 | 4 | 697 | 1 | 2 |

*2020*; *Wiecki et al., 2013*; *Zylberberg et al., 2016*) or neural response data (e.g., *Bollimunta et al., 2012*; *Brody and Hanks, 2016*; *Churchland et al., 2011*; *Ditterich, 2006*; *Genkin et al., 2021*; *Hanks et al., 2015*; *Howard et al., 2018*; *Latimer et al., 2015*; *Zoltowski et al., 2019*; *Zoltowski et al., 2020*), here, we seek to jointly model the relationship between accumulated evidence, choices, and neural activity.

The essence of our model is to describe a DDM-based accumulation process driven by sensory stimuli following *Brunton et al., 2013*, and relate the latent accumulation process to both neural responses and the rat's choice. Previous results have shown that this model is sufficiently flexible to accommodate the various behavioral strategies rats exhibit while performing this task (*Brunton et al., 2013*). The resulting model has a single latent variable, denoted $a(t)$, that evolves in time and represents the current, inner mental representation of the evidence in support of a left or right choice at each moment in time. This latent variable is shared by the neurons within a region (except where explicitly noted), so that each neuron's time-varying firing rate is a function of $a(t)$ on each trial. The key distinction of our approach is that the accumulator variable $a(t)$ drives both choices and neural activity, as described below.

Formally, the temporal evolution of the latent evidence $a(t)$ is governed by:

$$da = \lambda a dt + \Delta(t)\, dt + \sigma_a dW + \sigma_s \Sigma(t)\, \eta dt, \tag{1}$$

where $da$ is the amount $a(t)$ changes in a time $dt$. $\lambda$ is a leak parameter. $\Delta(t)$ and $\Sigma(t)$ indicate the difference and sum, respectively, of the number of right and left sensory clicks at time $t$, after the magnitude of the clicks has been adapted based on recent stimulus history (see parameters governing adaptation below, and Methods for additional details). $\sigma_a dW$ is a diffusive Gaussian noise process (or Weiner process) with scaling $\sigma_a$. $\sigma_s \Sigma(t)\eta$ is additive Gaussian noise induced by each click input, where $\sigma_s \Sigma(t)$ is the standard deviation of the click noise and $\eta$ is a Gaussian random variable with a mean of 0 and standard deviation 1.

If $a(t)$ becomes greater in magnitude than a symmetric boundary with magnitude $B$ (*Figure 1B*, dotted lines), then $da = 0$, and accumulation ceases for the remainder of the trial. To illustrate, the blue trajectory in *Figure 1B* crosses the boundary $B$ roughly one-third of the way through the trial, and thus remains constant thereafter.

The four terms of *Equation 1* each account for specific ways $a(t)$ might reflect accumulated evidence. The first two terms are designed to account for deterministic (non-random) dynamics exhibited by $a(t)$. The first term specifies how recent values of $a(t)$ influence future values and is governed by $\lambda$ that determines the timescale of this effect. Positive values of $\lambda$ correspond to unstable dynamics so that $a(t)$ grows exponentially. In this setting, early clicks have greater influence on $a(t)$ than recent clicks, because their impact grows with time. By contrast, negative values of $\lambda$ correspond to leaky

dynamics. In this setting, early clicks have a weaker influence on $a(t)$ than recent clicks because the impact of early clicks decays with time. When $\lambda$ equals zero, the sensory clicks are perfectly integrated. Previous results have shown that rats exhibit a range of accumulation strategies spanning these values of $\lambda$ (**Brunton et al., 2013**). The second term, $\Delta(t)dt$, specifies how the click stimulus is incorporated into $a(t)$. Because the task requires reporting whether there was a greater number of left or right clicks, only the total click difference is required to correctly perform it.

To account for stochasticity in the accumulation dynamics, the model also contains two forms of noise in $a(t)$. The first noise term, $\sigma_a dW$, corresponds to diffusive noise that corrupts $a(t)$ continuously in time. The final term, $\sigma_s \Sigma(t)\eta dt$, introduces noise into $a(t)$ that is proportional to the total number of clicks that occur at a given moment. The sum of clicks $\Sigma(t)$ is included so that the magnitude of the noise increases depending on the number of sensory clicks experienced at time $t$. **Figure 1B** illustrates the effects of these two noise terms: although the sensory inputs and leak are identical for both blue and black trajectories of $a(t)$, differences in noise lead the two trajectories to diverge so that one hits the boundary $+B$ while the other remains sub-threshold and continues to integrate the sensory stimulus.

To model animal choices, we assume that the accumulation variable $a(t)$ directly governs the animal's choice on each trial. Specifically, we describe the probability of a rightward choice as depending on $a(T)$, the accumulated evidence at the end of the stimulus period $T$, using a step function with 'lapses'. With probability $\gamma$ the animal picks one of the two sides without considering the stimulus, referred to as a 'lapse'. With probability $(1 - \gamma)$ the animal does not lapse, and makes a rightward choice if $a(T) > c$ and a leftward choice if $a(T) < c$, where $c$ denotes the choice criterion. This model can be expressed as:

$$P\left(d = R\right) = \gamma/2 + \left(1 - \gamma\right) H\left(a_T - c\right) \tag{2}$$

where $d \in \{L, R\}$ is the decision variable and $H(\cdot)$ is the Heaviside step function. As described above, when $a(t)$ crosses the decision bound $B$ a choice commitment is made, either to the left or the right, and no further evidence accumulation occurs. Previous work has found that parameterizing choice this way creates a model that is sufficiently flexible to describe animals' choice (**Brunton et al., 2013**) while remaining as simple as possible.

To model spike train data, we describe the time-varying firing rate of each neuron as a soft rectified linear function of the same accumulated evidence variable $a(t)$:

$$f_{\theta_n}(a(t)) = \text{softplus}(\theta_n a(t) + \theta_n^0(t)), \tag{3}$$

where $n$ indexes neurons, the softplus function (**Figure 1B**) is given by softplus$(x)=\log(1+\exp(x))$, and $\theta_n$ denotes the slope of the linear relationship between $a(t)$ and neuron $n$'s firing rate. The slope parameter, $\theta_n$, is fit separately for each neuron. A time-varying offset, $\theta_n^0(t)$, is included to capture time-varying changes in firing rate that do not depend on $a(t)$ (see Methods). The spikes of each neuron are modeled as a Poisson process with a time-dependent conditional intensity function $f_{\theta_n}(a(t))$. The softplus function (smooth rectified linear function) was used to ensure the expected firing rate was positive, and was selected because it is the simplest function to achieve this goal, and also based on prior success in similar studies (e.g., **Latimer et al., 2015**).

We refer to the set of all parameters that govern $a(t)$ and its relationship to the neural activity and choice data as $\Theta = \{\sigma_i, B, \lambda, \sigma_a, \sigma_s, \phi, \tau_\phi, \theta_{1:N}, c, \gamma\}$, where $\sigma_i$ is the variance of $a(t)$ at the start of the trial, and $\varphi$ and $\tau_\phi$ determine how the magnitude of each click is adapted based on the timing of recent clicks (see Methods). We fit $\Theta$ separately for each brain region using maximum likelihood (see Methods). Maximizing the likelihood of the data requires computing the temporal evolution of the probability distribution of $a(t)$ over the duration of a single trial, for all trials, and computing the probability of the observed spikes and choices under this distribution. The dynamics of this probability distribution can be expressed using the Fokker-Planck equation, and previous work has developed methods for numerically solving it (**Brunton et al., 2013**; see Methods). We refer to the value of $\Theta$ that maximizes the likelihood of the data as $\hat{\Theta}$. We verified that our method was able to recover the parameters that generated synthetic physiologically relevant spiking and choices data (**Figure 1— figure supplement 1**), and that parameter recovery was robust across a range of parameter values (**Figure 1—figure supplement 2**).

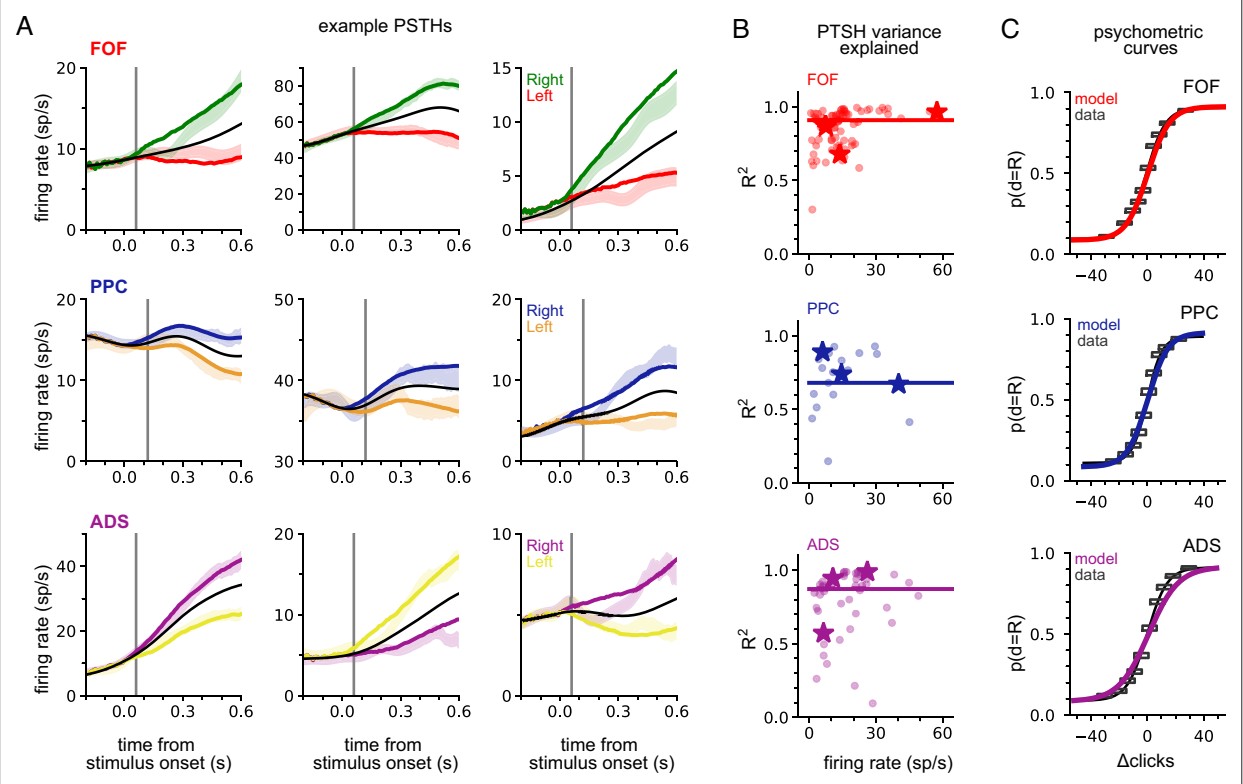

**Figure 2.** A shared accumulator model captures neural response and choice for each brain region. (**A**) Peri-stimulus time histograms (PSTHs) of three example neurons for each brain region (each row; frontal orienting fields [FOF]: red/green, posterior parietal cortex [PPC]: blue/orange, anterior-dorsal striatum [ADS]: purple/yellow). Spike trains were binned, filtered with a Gaussian kernel (std = 50 ms), grouped based on the strength of evidence, and averaged. Transparent shaded regions are ±1 standard error of the mean for the empirical data for each grouping. Colored curves are the mean of synthetic data simulated from the model with the parameters that maximize the likelihood of the data, grouped in a similar fashion. The black curve shows the trial-averaged firing rate, for all evidence strengths. Gray vertical lines indicate the average delay between the stimulus and the response for each brain region (see Methods). (**B**) Coefficient of determination ($R^2$) between empirical PSTH and synthetic data PSTH, for each neuron in each brain region. The data are plotted as a function of average firing rate. The median across the population is shown as a line. Points indicated with a 'star' refer to the data plotted in (**A**). (**C**) Probability of making a rightward choice as a function of cumulative difference in the number of clicks (psychometric curves) for empirical data (black lines) and data simulated from the model with the best fitting parameters (colored curves; FOF: red, PPC: blue, ADS: purple). Each curve is the curve of best fit, as computed by logistic regression. Rectangles indicate 25-th and 75-th quantiles of the data.

## Shared accumulator model captures neural responses and choices

We fit the model separately to data from each brain region. To verify model fits were consistent with data, we compared the peri-stimulus time histograms (PSTHs; *Figure 2A and B*) and psychometric curves (*Figure 2C*) of the empirical data to synthetic data simulated from the fitted model for each brain region. The PSTH of most neurons showed a characteristic choice preference that increased over time, consistent with accumulation. The model was able to capture this (*Figure 2A*). The model provided an accurate account of mean responses in all three brain areas (*Figure 2B*), with a median $R^2$ of 0.91, 0.68, and 0.87 for the FOF, PPC, and ADS, respectively (*Figure 2B*, colored lines). *Figure 2C* shows a comparison between true psychometric curves and the psychometric curve of the fitted model, confirming that the model also accounted for psychophysical choice behavior ($R^2$: 0.99—FOF; 0.99—PPC; ADS—0.97; see Methods for details). These analyses confirm that a shared accumulator model for each brain region is sufficient to capture the animals' choice sensitivity to the stimulus and strength of accumulated evidence reflected in each neuron's response.

## Different regions reflect different accumulator models, which all differ from model describing behavior

The primary motivation of our study was to learn accumulator models that incorporate precise stimulus-timing information and describe the animal's choices and temporally structured neural

activity. Previous efforts only modeled choices using stimulus-timing information (*Brunton et al., 2013*) or modeled neural activity without choices for tasks without detailed stimulus-timing information (*Latimer et al., 2015*; *Zoltowski et al., 2019*). We refer to our model that describes both neural activity and choices as the 'joint neural-behavioral model' or the 'joint model'. We compared the joint neural-behavioral model to a model where only the stimulus is used to model the animal's choice (i.e., neural activity is not used). To fit such a 'choice-only' accumulator model, we fit the same latent variable model using only choice data (see Methods).

*Figure 3A* shows the maximum likelihood parameters for the joint and choice-only accumulator models for each brain region. Neural data was not used for the choice model so brain region designates the cohort of animals from which the choice data was taken. We stress that because of this, each fitted choice model uses different behavioral choice data, and thus the fitted parameters vary from fitted model to fitted model. Both fitted models exhibited strong adaptation ($\phi << 1$) consistent with prior work fitting choice accumulator models (*Brunton et al., 2013*). This indicates that a stimulus pulse that occurs in rapid succession following other pulses has a smaller effect on $a(t)$ than an isolated pulse. Each model was impacted by different forms of noise: choice models exhibited small diffusive noise ($\sigma_a \approx 0$) and large stimulus noise ($\sigma_s >> 1$), consistent with earlier findings, while joint models exhibited large diffusive noise ($\sigma_a > 0$) and large initial variability in $a(t)$ ($\sigma_i >> 0$). The effect of these different parameters can be seen in *Figure 3B*: choice models have smaller initial variance and more variability when clicks arrive, while joint accumulator models have larger initial variance and diffusive noise. Large initial variance in the joint model likely reflects variability in neural responses prior to stimulus onset (*Churchland et al., 2010*). Strong accumulation noise in the joint model was also found when the negative binomial distribution, a more flexible observation model, was used, suggesting that this finding was not sensitive to the Poisson observation model (*Figure 3—figure supplement 1*). Differences in diffusive noise between the joint and choice-only models suggest that accumulation dynamics underlying neural activity is impacted by noise that is resolved at the level of a behavioral accumulator model.

We also compared the best-fit parameters across the three, separately fit, brain regions (*Figure 3A*). We focus on one of the most salient differences—the leak or instability parameter $\lambda$. Although there was no significant difference in the value of $\lambda$ across the cohorts of animals in the choice-only model, we found substantial differences across brain regions in the joint model fits (*Figure 3A*). The PPC and ADS data were best fit by leaky accumulator models ($\lambda < 0$). Surprisingly however, the FOF data was best described by a model with unstable accumulation dynamics ($\lambda > 0$) meaning that the model's accumulator (and thus firing rates) are more strongly affected by early stimulus clicks. The stronger weighting of earlier clicks was compounded further by the low accumulation bound of the model that best described FOF data. Such a low bound, in conjunction with unstable accumulation, causes $a(t)$ to stop evolving early in the trial (*Figure 3B*). This results in a phenomenon known as 'primacy encoding', in which early clicks more strongly impact the animal's choice while later clicks are ignored. We confirmed this finding in the FOF using a generalized linear model (GLM; see Methods and *Figure 3—figure supplement 2*). This result is consistent with previous work suggesting that the FOF has a categorical representation of $a(t)$ (*Hanks et al., 2015*). We expand on these findings in light of other studies of the FOF in the Discussion. Collectively, these results indicate that all three brain regions were best described by accumulator models that differed in their best fitting parameters (and thus exhibit dramatically different accumulation dynamics), and that each region's data was likewise best described by a model that differed from that which best described accumulation at the level of the animal's choice.

## ADS is better described by multiple, independent accumulators

Our model describes the spiking activity of a population of simultaneously recorded neurons as relying on a single shared latent variable. To assess whether this is indeed the best description of the data, we compared it to an 'independent-noise accumulator model' where each neuron is driven by an accumulator with its own independent noise (*Figure 4A*; Methods). It is worth emphasizing that the independent-noise model is identical to the shared-noise model in the way it is parameterized (i.e., number and form of the model parameters) but only differs in the structure of the latent accumulation noise. If trial-to-trial spiking covariation is produced by temporal covariation in the accumulator due to noise, the independent-noise model (which does not share this covariation) should not account

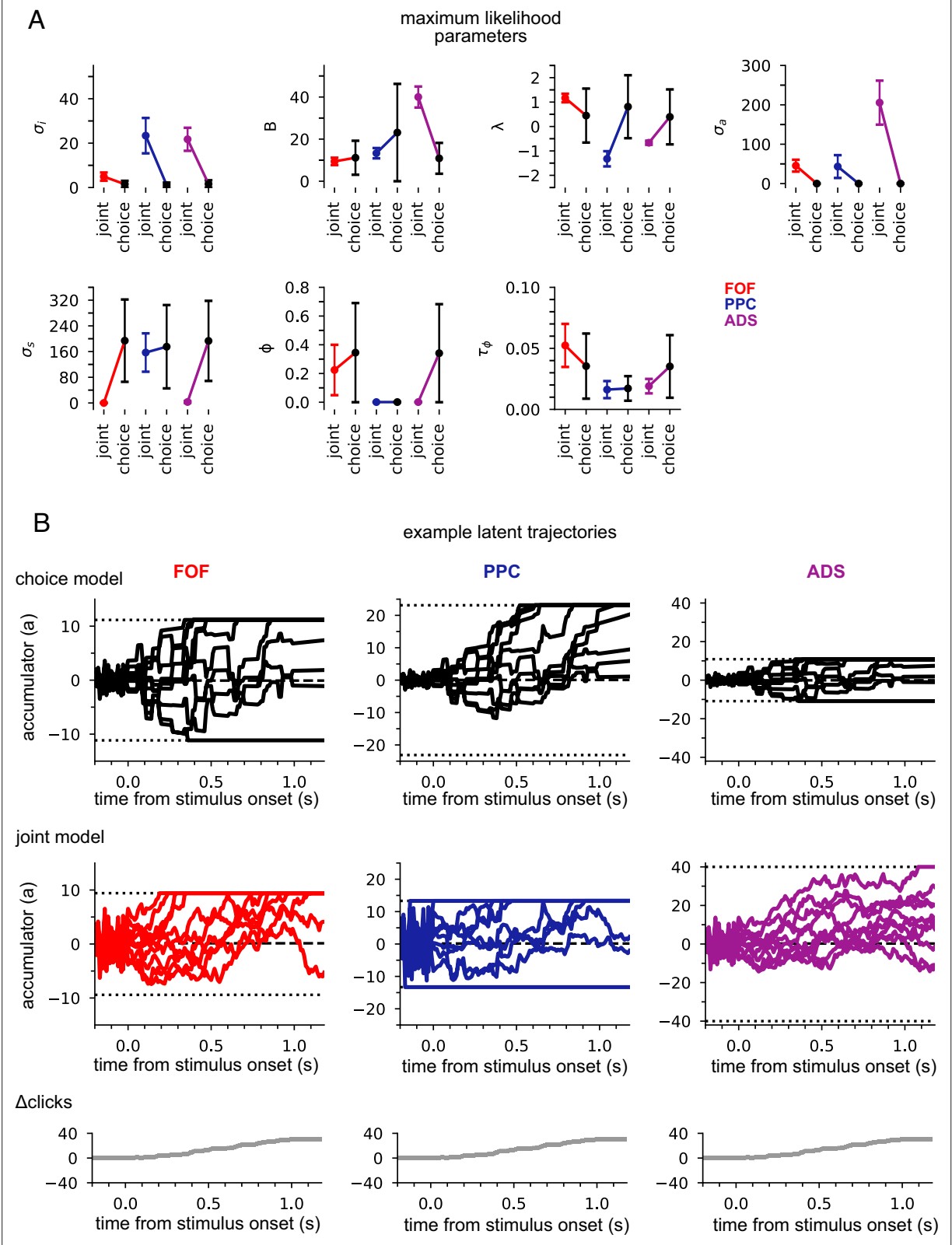

**Figure 3.** Data from different regions is best fit by different accumulator models. (**A**) Maximum likelihood parameters that govern $a(t)$ for the joint neural-behavioral model and the choice-only model. Error bars, computed by the Laplace approximation (Methods), are ±2 standard deviations. Parameters are $\sigma_i$: initial variance, $B$: accumulation bound, $\lambda$: drift, $\sigma_a$: accumulation noise variance, $\sigma_s$: click noise variance, $\varphi$: adaptation strength, $\tau_\phi$: adaptation timescale. (**B**) 10 example trajectories with different noise instantiations for one trial for the choice model (top) and the joint model (middle)

*Figure 3 continued on next page*

*Figure 3 continued*

for each brain region, and cumulative sum of the click stimulus for each trial (bottom). The dotted black lines (top and middle) indicate the accumulation boundary value for each model.

The online version of this article includes the following figure supplement(s) for figure 3:

**Figure supplement 1.** Model comparison using Poisson or negative binomial observation model.

**Figure supplement 2.** Generalized linear model (GLM) analysis of individual sessions.

**Figure supplement 3.** Maximum likelihood parameters of joint model for each frontal orienting fields (FOF) rat individually.

**Figure supplement 4.** Comparison of maximum likelihood parameters for three models: joint (neural/choice) model, choice-only model, and independent-noise joint model, when fit to all data, or using cross-validation data.

for the data as well, suggesting that correlations in the data can be attributed to correlated diffusive noise reflected in the shared model. We fit the parameters of the independent-noise model using the same optimization method but with a different log-likelihood function (see Methods). Because the independent-noise model contained multiple accumulators (one for each neuron), the animal's choice was modeled differently than for the shared-noise model (see Methods). We focused on the FOF and ADS datasets because they contained a sufficient number of simultaneously recorded neurons to make this comparison (*Table 1*). The maximum likelihood parameters for the two models for both regions were similar (*Figure 4—figure supplement 1*), except for the initial accumulator variance parameter which differed significantly.

We used fivefold cross-validation to determine which model better described each dataset. Comparing the cross-validated log-likelihood, we found that the independent-noise model provided a better description of choices and neural activity from ADS, while the shared-noise model provided a slightly better description of FOF data (*Figure 4B*). This finding supports the conclusion that neural responses within the ADS reflect independent accumulation processes, while neurons in the FOF reflect a single latent accumulator. Although ADS datasets with four or more neurons provided the primary contribution to these results (*Figure 4—figure supplement 2A*), when the number of neurons in ADS datasets was subsampled to match the maximum number of neurons in FOF sessions (three neurons), the ADS recordings still favored an independent-noise accumulator model (*Figure 4—figure supplement 2B*). We fit the shared-noise and independent-noise model to neural data only (excluding choice data) and found consistent results (*Figure 4—figure supplement 2D*), suggesting this difference is not due to contributions from the animal's choice, which was modeled differently in each model (see above).

To further examine this result, we computed the 'shuffle-corrected' cross-correlation function (Methods; *Perkel et al., 1967*; *Smith and Kohn, 2008*) for all pairs of simultaneously recorded neurons to examine spiking covariation in the empirical data and synthetic data from the fit models (*Figure 4C and D*). To shuffle-correct, we took the raw cross-correlation and subtracted the cross-correlation of the PSTHs of two neurons (for left and for right trials separately). This provides a measure of the neurons' correlation beyond what is to be expected from the PSTHs (i.e., *Figure 2A*).

Synthetic data of both models captured trends in the shuffle-corrected cross-correlation function at slower timescales but failed to capture fluctuations on short timescales. Across all pairs of simultaneously recorded neurons (70 pairs in total), we found that the shared and independent-noise accumulator models provided approximately equally accurate fits to the shuffle-corrected cross-correlations (mean $r$ of 0.55 for shared model and 0.57 for independent-noise model for FOF; 0.63 for shared model and 0.60 for independent-noise model for ADS). This shows that both models capture correlations in trial-to-trial neural responses beyond those accounted for by the PSTH. These correlations likely arise from trial-to-trial differences in the exact sequence of clicks, which are not reflected in the PSTH for left- or right-choice trials. Although FOF weakly favored a shared-noise model and ADS favored an independent-noise model (*Figure 4B*) the comparable ability for each model to capture the shuffle-corrected cross-correlation function for each region suggests that these correlations are primarily stimulus-induced and not a manifestation of non-stimulus-induced (i.e., 'noise') correlations, which are weak if present at all. Although these results suggest that each model fits the data equally well, the results of *Figure 4B* suggest that the independent-noise model may be accounting for intricate features of the ADS data not reflected in the shuffle-corrected cross-correlation function.

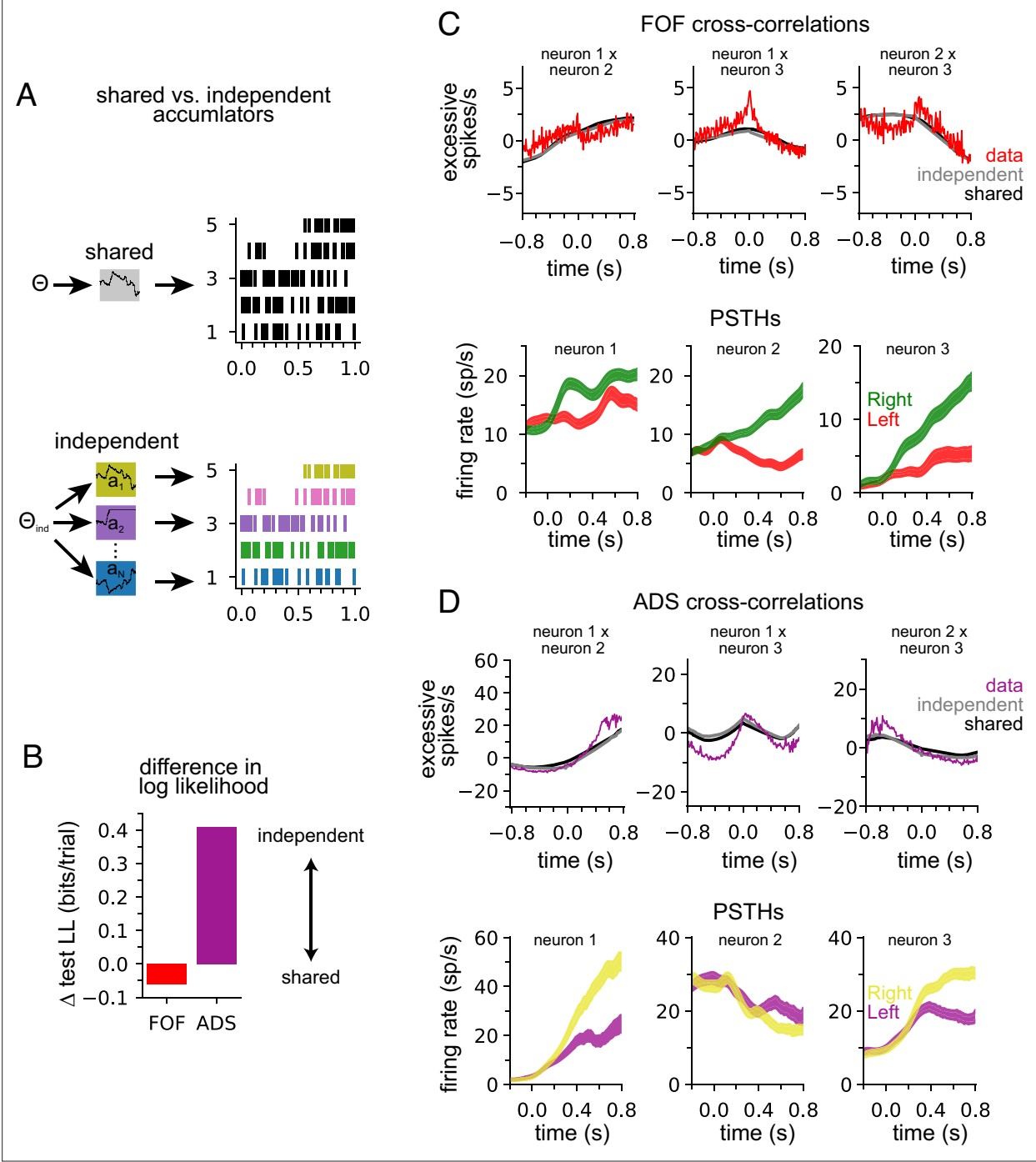

**Figure 4.** Anterior-dorsal striatum (ADS) is better described by independent accumulators. (**A**) For the shared-noise accumulator model (top), a set of parameters defines the dynamics of a single accumulator, which drives the spiking activity of the entire population. In the independent-noise accumulator model, a set of parameters defines the dynamics of an ensemble of independent accumulator models, which each individually determine the spiking of a single neuron. (**B**) Difference in test log-likelihood (bits/trial) for the shared-noise vs. independent-noise accumulator models. (**C**) Empirical (red) and synthetic (shared: black; independent; gray) shuffle-corrected cross-correlation function for three simultaneously recorded neurons from the frontal orienting fields (FOF). Corresponding peri-stimulus time histograms (PSTHs) are shown below for reference. (**D**) Same as (**C**) for three (of five) simultaneously recorded neurons from the ADS.

The online version of this article includes the following figure supplement(s) for figure 4:

**Figure supplement 1.** Maximum likelihood parameters for the joint (neural/choice, i.e., shared-noise) model and independent ('ind.') noise joint model.

**Figure supplement 2.** $\Delta LL$ between the shared-noise and independent-noise accumulator model.

To validate that neural responses in the ADS weakly covary, as suggested by an independent-noise model, we computed a measure of response dimensionality known as the participation ratio (*Litwin-Kumar et al., 2017*). The participation ratio is computed using the eigenvalues of the covariance matrix of firing rates (Methods). If all firing rates are independent the eigenvalues will all be equal and the participation ratio will equal the number of neurons. If the firing rates are correlated such that some eigenvalues are small (or perhaps zero) the participation ratio will reflect this and the dimensionality of the data will be less than the number of neurons. Consistent with our modeling results, we found that responses in ADS had higher dimensionality than in FOF (i.e., ADS exhibited less firing rate covariation) and that ADS sessions with greater dimensionality were those that favored the independent-noise model (*Figure 4—figure supplement 2C*).

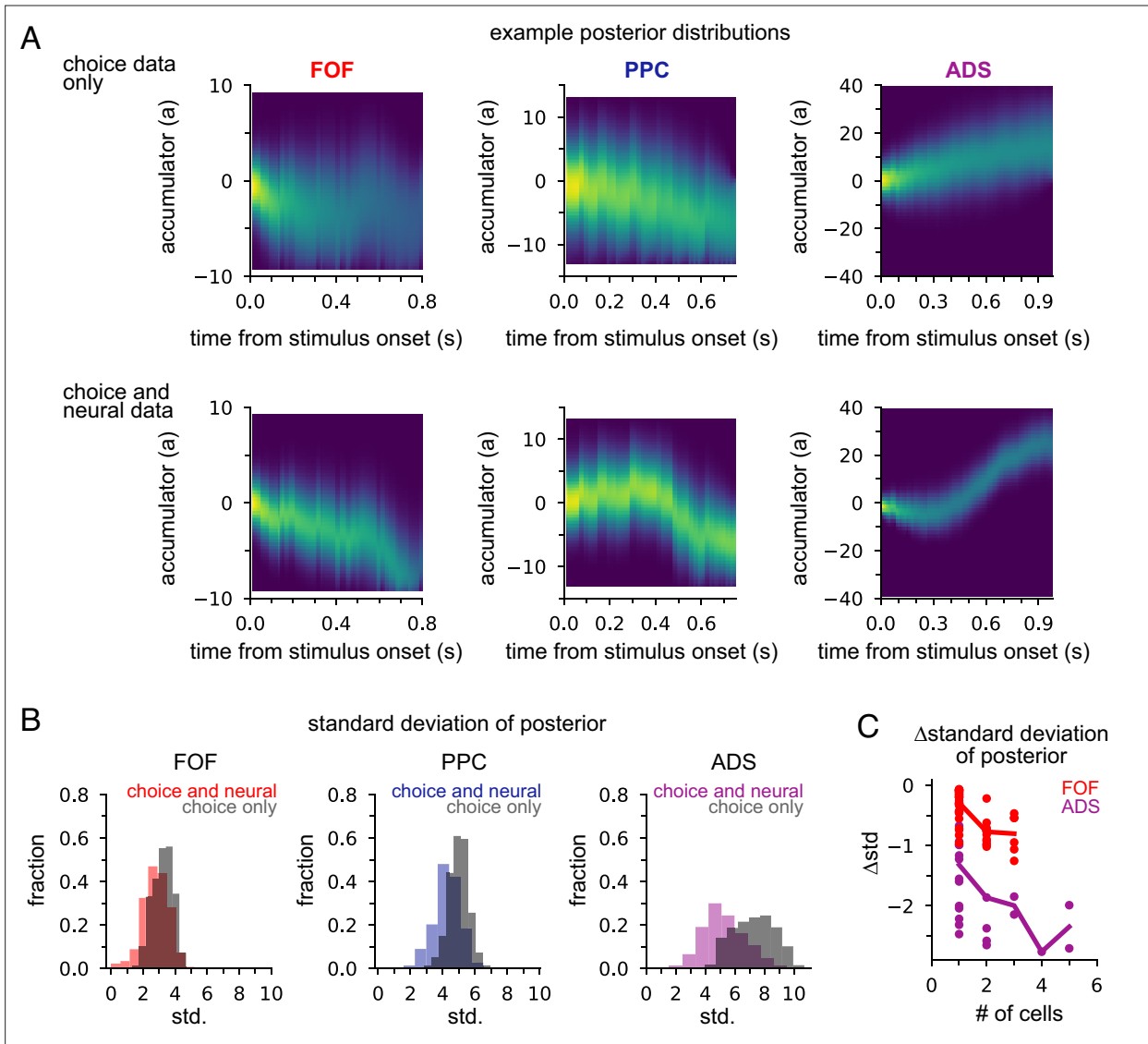

**Figure 5.** Neural data provides more information about accumulated evidence on single trials than choice alone. (**A**) Posterior distribution of *a*(*t*) under the joint model (excluding captured mass at the boundary) given only the choice (top row) and given spike times and choice (bottom row), for a single example trial. Columns show example trials for different brain regions. (**B**) Histogram of joint model posterior standard deviations given choice data (*black*) or both neural and choice data (*colors*) for all three brain regions. (**C**) Difference in choice-conditioned joint posterior standard deviation and neural- and choice-conditioned joint posterior standard deviation as a function of the number of simultaneously recorded neurons. Each point is the difference in the average posterior standard deviation for a session. Negative values indicate that the neural- and choice-conditioned posterior had smaller average standard deviation than the choice-conditioned posterior.

## Neural data provides more information about accumulated evidence than choice

Next we examined how neural data affected inferences about accumulated evidence. We computed the posterior distribution over the accumulator variable $a(t)$ for the joint model, given choice data only, or given neural and choice data. The posterior distribution combines information from multiple sources—stimulus, choice, and neural activity—to offer a concise window into the animal's internal state of evidence accumulation. *Figure 5A* shows the posterior distribution for three example trials (one for each brain region) when only choice data was included and when both choice and neural data were included. The choice data posterior was broad; a large set of $a(t)$ trajectories were all consistent with the animal's choice. However, when we considered both choice and neural spiking activity, we obtained a substantially narrower distribution over $a(t)$, meaning including neural data in the joint model offers greater confidence in the precise value of accumulated evidence at each moment within a trial.

To quantify this difference, we computed the standard deviation of the two posteriors (*Figure 5B*). For all brain regions, the median posterior standard deviation given neural data and choice was substantially smaller than when conditioning only on choice (*Figure 5B*; median difference FOF: 0.46; PPC: 0.72; ADS: 2.23). This reduction in the posterior width increased with the number of neurons (*Figure 5C*). The increased certainty about $a(t)$ provided by neural activity makes intuitive sense: temporally specific spiking activity (e.g., in the middle of a trial) allows one to infer that $a(t)$ has increased in favor of a choice, whereas choice information can only offer certainty about the range of $a(t)$ at the end of the trial.

## Joint neural-behavioral model improves choice decoding

We designed our joint model with the expectation that combining choice data, neural responses, and stimulus information within an accumulation framework would lead to greater insight into decision-making than models that lacked these features. We tested this expectation by comparing choice decoding accuracy of the joint model on single trials to models that used stimulus information and only choice data or only neural data (see Methods). We found that choices could be predicted more accurately under the joint model, which took into account the stimulus, neural activity, and choices, than under the choice model, which used stimulus information and choices alone. We quantified this improvement in test log-likelihood and percent correct (*Figure 6A*). The joint model had higher test log-likelihood for choice data and choice-prediction accuracy for all three brain regions, with the joint model of FOF data showing an almost 50% improvement in test log-likelihood and a 6% increase in prediction accuracy. The posterior mean of the joint model and the posterior mean of the choice model is shown in *Figure 6B* for three example trials. In all examples, the joint model correctly predicted the choice the animal made (indicated by the arrow), whereas the choice-only model failed because its prediction was based on the stimulus. This increased performance derives from the choice-informative spiking information contained in the posterior (*Figure 5*) that the choice model lacks.

If neural activity is highly correlated with the motor report (e.g., activity from motor neurons controlling orientation), we would expect the neural activity to be a good predictor of the animal's choice. In such a case, a model that predicted choice without the framework of the DDM accumulator but using neural activity would have high accuracy. We compared our accumulator-based joint model to a logistic regression model (i.e., Bernoulli GLM) which used the final accumulated click difference and the trial-summed spike count for each neuron as regressors (Methods). Decoding under the joint accumulator model significantly outperformed logistic regression (*Figure 6A*, *GLM*). The performance of the GLM did not depend strongly on the time window considered: decoding of choice using spikes from the last 50 ms (*Figure 6A*, *GLM 50 ms*), 100 ms, 150 ms, 200 ms, and 250 ms before a decision all performed similarly (*Figure 6—figure supplement 1*). This shows that the joint accumulation framework and the fine timescale dynamics of the joint model captures features of the spike trains that are useful for predicting the animal's choice, above and beyond the information carried by spike counts in particular time windows before the choice.

## Putative changes of mind are common in ADS, rare in FOF

The previous analysis illustrated how the joint accumulation framework, combined with temporally precise neural responses, can accurately predict animal choices. Numerous studies have shown that

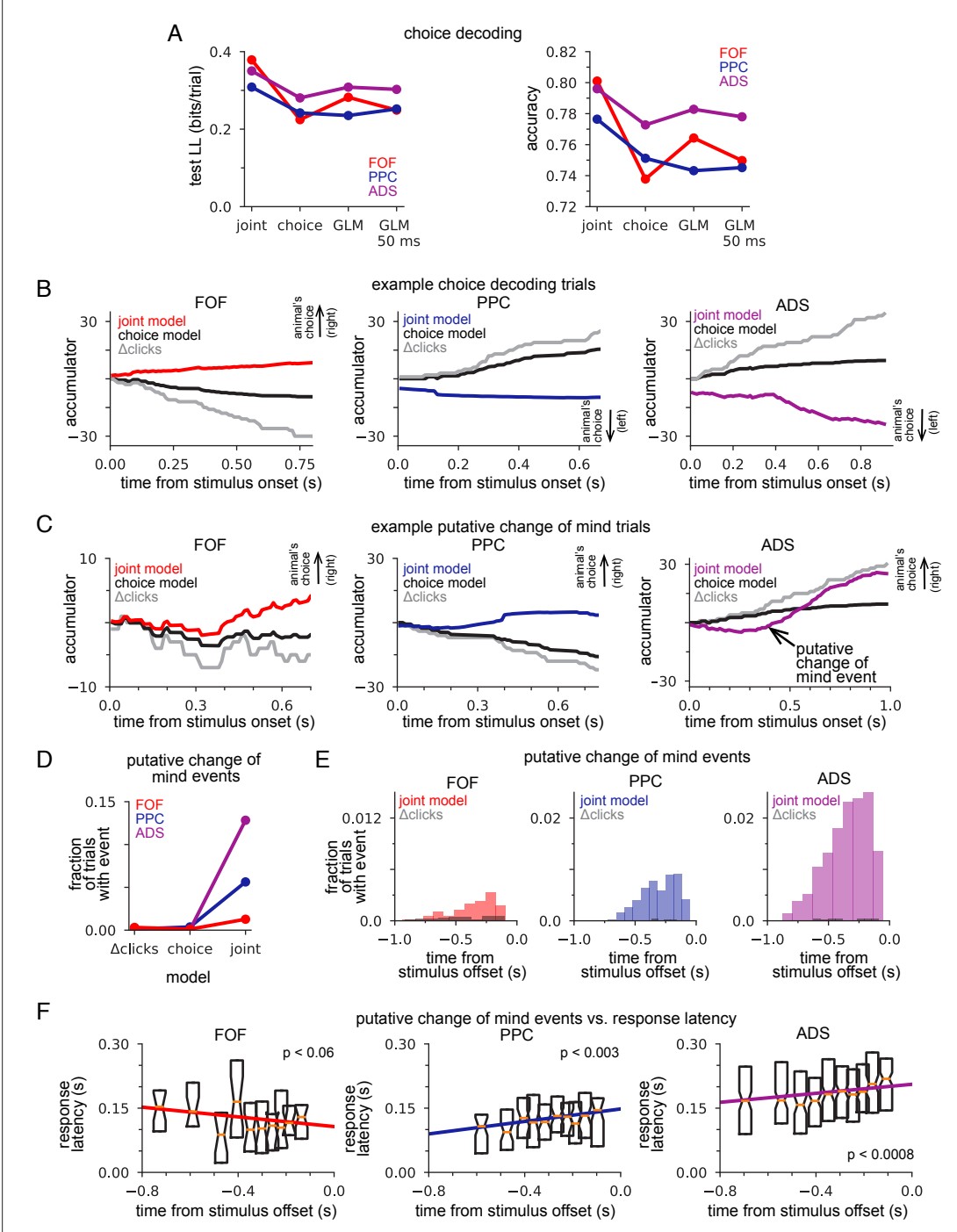

**Figure 6.** Joint neural-behavioral model improves choice decoding. (**A**) Choice-prediction accuracy, quantified with log-likelihood (left) and percent correct (right) on test choice data for four models: joint neural-behavioral model, choice-only model, and two logistic regression models (Methods). Values greater than zero indicate that the model can predict choices better than a baseline model that only knows the marginal probability of a rightward choice. (**B**) Posterior mean of $a(t)$ conditioned on the neural activity for the joint model (colors), the distribution of $a(t)$ for the choice-only model (black), and the cumulative click difference (gray) for three example trials (one for each brain region). 'Animal's choice' arrow indicates the choice (left or right) the animal made on that trial. (**C**) Putative change of mind events, where the posterior mean of the joint model crossed the decision threshold. The corresponding distribution of $a(t)$ for the choice-only model (black) and the cumulative click difference (gray) for the same trial are shown for comparison. 'Animal's choice' arrow indicates the choice (left or right) the animal made on that trial. (**D**) Fraction of trials that contain at least one putative change of mind event for the cumulative click difference, the choice model, and the joint model, for each brain region. (**E**) Fraction of trials for which a putative change of mind event occurs at the specified time relative to the end of the stimulus for the joint model (color) and the cumulative click difference (black) for each brain region. (**F**) Choice response latency as a function of timing of putative change of mind events relative to stimulus offset

*Figure 6 continued on next page*

*Figure 6 continued*

for each brain region. Bar plots show the 25–75 percentiles of the choice response latency for putative change of mind events occurring at similar times. The colored lines indicate the line of best fit for each brain region computed by linear regression.

The online version of this article includes the following figure supplement(s) for figure 6:

**Figure supplement 1.** Generalized linear model (GLM) choice decoding (as in *Figure 6A*) using spikes in different time windows relative to stimulus offset.

**Figure supplement 2.** Accuracy on putative change of mind event trials and non-event trials.

subjects making decisions based on noisy stimuli will vacillate before reporting a decision (*Kaufman et al., 2015*; *Kiani et al., 2014*; *Resulaj et al., 2009*). Switches of a subject's provisional decision have been referred to as 'changes of mind' (*Boyd-Meredith et al., 2022*; *Peixoto et al., 2021*). We used our joint accumulator model to identify putative changes of mind from our neural recordings, to examine how decision commitment is manifested in different brain regions. We examined the temporal dynamics of the joint model posterior, conditioned on neural activity only, to find putative changes of mind: moments when posterior mean crossed from one side of the decision threshold to the other. We required that the conditioned posterior mean remained on one side of the decision threshold for at least 50 ms before and after the crossing and achieved an absolute magnitude greater than 2 during that 100 ms window (see Methods).

*Figure 6C* shows three example putative change of mind trials. We also plot the posterior mean of the choice model (black) and the cumulative click difference (gray) for comparison. In all three examples, the joint model posterior mean crossed the decision threshold, ending on the side corresponding to the animal's choice. Sign changes in the cumulative click difference were rare, as were putative change of mind events under the choice-only model, both of which could only be caused by the stimulus (*Figure 6D*). In contrast, putative change of mind events were observed frequently under the joint model for all three brain regions (*Figure 6D*). This shows that putative change of mind events reflect information about the accumulator carried in neural activity. Putative change of mind events were observed least frequently in the FOF and most frequently in the ADS (*Figure 6D*); compounded by our initial finding, that different brain regions are best fit by different accumulator models (*Figure 3*), these results further support the view that the decision-making dynamics in each brain region are fundamentally and consequentially different.

The animal's performance improved on putative change of mind event trials (fraction correct: FOF: 0.88 vs 0.74; PPC: 0.87 vs 0.74; ADS: 0.85 vs 0.76; *Figure 6—figure supplement 2A*) and the choice prediction of the joint model was also more accurate (fraction correct: FOF: 0.92 vs 0.80; PPC: 0.88 vs 0.77; ADS: 0.88 vs 0.78; *Figure 6—figure supplement 2B*), suggesting that the decision-making dynamics that give rise to these events primarily correct incorrect decision-making dynamics early within a trial. Initial variability in the accumulation dynamics, as reflected in neural responses, was found to be greater in both PPC and ADS (*Figure 3A*), regions for which putative changes of mind were more likely (*Figure 6D*), consistent with this assumption. Furthermore, putative change of mind events were more likely to occur at later moments in the trial, usually not long before the stimulus ended (*Figure 6E*), consistent with the assumption that they generally correct incorrect early-trial dynamics. To more firmly connect putative change of mind events to the animal's behavior, we performed linear regression to compare the time of the event relative to the end of the stimulus to the response latency (*Figure 6F*). We found a statistically significant effect for the PPC and the ADS (PPC: $p < 0.003$; ADS: $p < 0.0008$; two-sided t-test), which both showed a slower response time when a change of mind event occurred closer to the end of the stimulus. These results illustrate the potential of our framework for uncovering putative covert changes of mind within neural activity, and demonstrate the varying way in which decision-making dynamics—both prior to stimulus onset and during the stimulus period—differ in different brain regions.

## Discussion

We developed a probabilistic latent process model to simultaneously describe neural activity and choices during an evidence accumulation decision-making task. We fit the model to data from three brain regions and found that the dynamics of accumulation that best-fit choices and neural data from each brain region differed significantly across brain regions, and from the accumulation model that

best described the animal's choices. We found that including neural activity in the model provided rich, moment-by-moment information about the animal's choice. The inferred accumulation model could be used to examine estimates of the animal's moment-by-moment provisional choice, and by doing so, we found differing choice-related dynamics in each brain region, dynamics that meaningfully related to other measures of behavior such as reaction time. Collectively, our results argue for the existence of very different accumulation dynamics in different brain regions, dynamics which each differ greatly from the dynamics giving rise to behavior. An exciting future application of our modeling framework is to model multiple, independent accumulators in several brain regions which collectively give rise to the animal's behavior. Such a model would provide incredible insight into how the brain collectively gives rise to behavioral choices.

There has been substantial work relating neural activity to evidence accumulation. The logic underlying this work (e.g., *Churchland et al., 2011*; *Gold and Shadlen, 2007*; *Hanks et al., 2015*; *Mante et al., 2013*; *Ratcliff et al., 2003*; *Yartsev et al., 2018*) is that behavior is well approximated by gradual evidence accumulation (*Ratcliff and McKoon, 2008*). Numerous studies have probed whether neurons in any given brain are involved in encoding or computing a correlate of this behavior-level evidence accumulation. A rarely emphasized assumption is that the accumulation process, at the level of individual brain regions, will be similar to the accumulation process at the level of the organism's behavior. This assumption need not be correct. As in the example mentioned in the Introduction, two brain regions, one representing a leaky accumulator from which recent evidence is best decoded, and another representing an unstable accumulator from which the earliest evidence is best decoded, could combine to generate behavior that is well described by stable evidence accumulation, in which evidence from throughout behavioral trials is weighted approximately equally. One should not conclude that neural activity best explained by a leaky or by an unstable accumulator is unrelated to behavior that is best explained by stable accumulation. Other properties, in addition to leakiness/instability, may also differ across contributing brain regions. Developing a formal approach to fit the parameters of evidence accumulation models from neural data as well as from choices provided us with the opportunity to probe this assumption. Our results suggest that it is *not* correct. Elucidating the neural basis of evidence accumulation for decision-making may require understanding how brain regions with neural activity that appears driven by accumulators with potentially very different properties combine, and perhaps counterbalance each other, so as to produce the organism's behavior.

Our approach extends and complements existing approaches that construct formal mathematical models of decision-making which combine both behavioral data and neural data. These models leverage both neural and behavioral observations to jointly infer decision-making parameters, as we've done here (see *Turner et al., 2019* for a comprehensive overview). However, the majority of these approaches have tended to emerge from the field of cognitive neuroscience, and as such, have predominantly focused on models for application to neural data acquired by other methods, such as EEG, fMRI, etc. (e.g., *Turner et al., 2015*; but also see *Frank et al., 2015*). Our approach adds to these efforts by offering a method that can combine fine timescale single-unit recordings with behavioral measurements specifically during pulse-based evidence accumulation tasks, thereby offering a moment-by-moment picture into the latent dynamics that underlies cognition. Continued development of joint models such as our approach and existing approaches in the field of cognitive neuroscience are critical to quantitatively understand the latent processes underlying cognition.

One of our most surprising discoveries was that neural data from the FOF was best modeled by an accumulator consistent with a 'primacy' strategy in which early stimulus clicks have an outsize impact on neural activity and choice compared to later clicks. Coupled with the low accumulation bound of the model fit to the FOF, our analysis suggests a model of FOF accumulation where a subject prematurely commits to a decision based on early sensory evidence. Previous analysis of these data did not find that FOF activity was described by an unstable accumulator because the accumulator model was not learned from neural activity, only choices (*Hanks et al., 2015*). This prior analysis identified an alternative interpretation of FOF activity: FOF activity exhibited a step-like encoding of accumulated evidence that was unbounded, consistent with the FOF encoding a categorical representation of choice (*Hanks et al., 2015*). At a strategic level, this interpretation is consistent with the model of FOF activity we identified. Noting that, in this task, the stimulus will rarely cause the accumulator to switch sign (*Figure 6D*), a step-like encoding of an unbounded accumulator that does not switch sign will appear very much like an bounded accumulator: for either model, the accumulator will quickly jump to

its largest value and remain there. Additional experiments and modeling are required to differentiate these two models.

A primacy encoding model of the FOF is supported by our change of mind analysis. Putative change of mind events identified from neural activity occurred less frequently in the FOF than other regions (p<4.5694e-82 FOF vs. PPC; p<3.4585e-323 FOF vs. ADS; Fisher's exact test) consistent with an early-commitment strategy in the FOF. A recent study of the FOF during an accumulation task in which evidence dynamically changed throughout a trial found that FOF activity reflected evidence across stimulus-induced 'overt' changes of mind, and that these events were common in the FOF (**Boyd-Meredith et al., 2022**). It's important to note that we likewise found that FOF reflects evidence across changes of mind, but we identified rarely occurring non-stimulus-induced 'covert' changes of mind during a task in which the evidence was static, and thus our results do not conflict with those findings.

A primacy encoding model of the FOF is also both supported by and offers context to prior FOF inactivation studies (**Erlich et al., 2015**). Behavioral modeling of choices in conjunction with bilateral pharmacological inactivation found that FOF inactivation led to leakier accumulation when producing choices (**Erlich et al., 2015**). Leakier accumulation at the level of choice also implies that later stimulus information disproportionately impacts choice, precisely the impact predicted if an early stimulus favoring brain region, such as the FOF, was silenced. A more complete model relating accumulation dynamics in multiple brain regions to choice-related accumulation dynamics at the level of behavior would aid in understanding how silencing individual brain regions, with their region-specific accumulation dynamics, impacts accumulation at the level of behavior.

Our novel change of mind analysis identified both the ADS and PPC as regions that showed frequent instances of choice vacillation during this task. Prior studies in related tasks found that neural responses in one of these regions, the PPC (or its primate homolog), reflect information related to already experienced trials (**Akrami et al., 2018**; **Purcell and Kiani, 2016**), consistent with our interpretation of prestimulus neural responses being suboptimally tuned for the upcoming trial and thus requiring mid-trial correction. Given the large initial accumulator variance of ADS and the presence of frequent putative change of mind events in this region, activity in ADS seems poised to also reflect these types of trial history-dependent responses as well. Future experiments and analysis are required to determine this.

Previous studies that fit this model to only choices developed specific interpretations of the accumulation strategy used by animals (**Brunton et al., 2013**). One difference between choice accumulator models and joint neural-behavior models is the differential impact of accumulator noise vs. stimulus noise. Choice-only models have typically indicated that stimulus noise is the primary cause of systematic behavioral uncertainty (**Brunton et al., 2013**), whereas our joint models suggest that this impact is weaker than diffusion noise. One interpretation of this difference is that at the level of a single neural population, diffusive noise plays a stronger role in producing uncertainty in $a(t)$ than stimulus noise, whereas at the level of the entire brain's encoding of accumulated evidence, this diffusive noise 'averages out' and residual stimulus noise remains. Understanding how multiple brain regions work together to produce a model of accumulated evidence at the level of behavior is an important future direction of this work.

Several extensions of our framework are readily apparent. Increasing the number of recorded neurons led to an improved estimate of $a(t)$. As the density of neural recordings increases (**Luo et al., 2020**), the explanatory power of our model will increase. Although we have extended the evidence accumulation model to include neural responses and choice, we could extend it further to describe additional physiological or behavioral variables (e.g., from annotated video data, pupil-dilation measurements, response time, etc.). Including these additional behavioral measures would further inform the inferred accumulator model, providing a clearer window into the internal factors governing choices. Although we considered a specific evidence accumulation model due to its normative interpretation, our framework can readily accept modifications and extensions of its dynamical equations (e.g., **Genkin et al., 2021**). More sophisticated (e.g., nonlinear) dynamics of accumulated evidence or more refined models of accumulation noise are two examples. Our framework can also accommodate more elaborate and/or appropriate relationships between accumulated evidence and neural responses, as we briefly explored by considering the negative binomial distribution (**Figure 3—figure supplement 1**). Changing this relationship would open the door to using this approach with other types

of data, such as imaging data. Although our framework was developed with the specific application to a pulse-based accumulation task in mind, it is not confined to this. Our framework can be adapted to any task where noisy temporal accumulation of evidence is thought to play a role, and for which neural recordings and behavioral choices reflect this process (*Aguillon-Rodriguez et al., 2021*). Finally, while a major motivation of our approach was to develop a framework for identifying a specific normative and mechanistic accumulation model, its rigidity makes it difficult to capture varying features present in the data. Extending the model to include additional latent processes alongside a rigid accumulation model (*Zoltowski et al., 2020*) would enable the model to simultaneously account for currently unexplained variance in the data while preserving the model's ability to account for variance with an accumulation model. Doing so may offer a clearer picture of the evidence accumulation process by sweeping away unrelated variance with a more flexible, but less interpretable, latent process model.

## Methods

### Latent variable model

We model accumulated evidence as a one-dimensional DDM with a symmetric absorbing boundary (*Brunton et al., 2013*). On a single behavioral trial, the evolution of the accumulated evidence, $a(t)$, is governed by

$$da = \lambda a dt + \sigma_a dW + \sigma_s dt \left( \eta' \delta_{t,t_R} C_R(t) - \eta' \delta_{t,t_L} C_L(t) \right) \tag{4}$$

where $\lambda$ is the inverse of the drift time constant. $\sigma_a dW$ is a Wiener process with scaling $\sigma_a$. $\sigma_s \eta'$ are Gaussian variables with variance $\sigma_s^2$ and mean 1. $\delta_{t,t_L}$ and $\delta_{t,t_R}$ are the timing of left and right pulses, respectively, and $C_L(t)$ and $C_R(t)$ are the magnitude that each left or right click, respectively, has at time $t$. The impact of each click is modulated by sensory adaptation, based on the following equation:

$$\frac{dC_\alpha}{dt} = \frac{1 - C_\alpha}{\tau_\phi} + (\phi - 1)\left( C_\alpha \delta_{t,t_\alpha} \right), \tag{5}$$

where $\alpha = \{L, R\}$. We define the difference of the adapted click magnitude at time $t$ as $\Delta(t) = \delta_{t,t_R} C_R(t) - \delta_{t,t_L} C_L(t)$ and the sum of the adapted click magnitude at time $t$ as $\Sigma(t) = \delta_{t,t_R} C_R(t) + \delta_{t,t_L} C_L(t)$. By doing so, we can express *Equation 4* as:

$$da = \lambda a dt + \Delta(t) dt + \sigma_a dW + \sigma_s \Sigma(t) \eta dt, \tag{6}$$

where $\eta$ is a standard Normal. An absorbing boundary, $B$, if crossed, prevents $a(t)$ from evolving further (i.e. $da = 0$ if $a(t) > B$). The initial state of $a(t)$ is distributed normally with mean 0 and variance $\sigma_i^2$. We refer to all parameters that govern the dynamics of $a(t)$ as $\theta_a = \{\sigma_i, \lambda, B, \sigma_a, \sigma_s, \phi, \tau_\phi\}$.

### Computing the distribution of the latent state

The temporal dynamics of the probability distribution of $a(t)$, $P(a(t))$, can be expressed as a Fokker-Planck equation,

$$\frac{\partial P(a(t))}{\partial t} = \frac{\sigma_a^2 + \sigma_s^2 \Sigma(t)}{2} \frac{\partial^2 P}{\partial a^2} - \frac{\partial \left( (\lambda a + \Delta(t)) P \right)}{\partial a} \tag{7}$$

We numerically compute the solution to *Equation 7* by dividing $P(a(t))$ into a set of $n$ discrete spatial bins, and determine how mass moves after a discrete temporal interval, $\Delta t$. The discrete time dynamics of $P(a_t)$ are Markov, and obey the following equation:

$$P(a_t) = M(\theta_a, \delta_t) P(a_{t-1}), \tag{8}$$

where $\delta_t$ is the collection of left and right clicks at time $t$. The transition matrix $M(\theta_a, \delta_t)$ is determined using methods established in *Brunton et al., 2013*. Briefly, for each spatial bin, the deterministic effect of the dynamics on the probability mass is computed, and this is convolved with a discrete approximation to a Gaussian distribution with the appropriate variance and a finer spatial resolution than the initial spatial resolution described above, to determine the various locations of that probability mass at the next time bin. Because the location of each bin of mass after the Gaussian convolution is not

likely to correspond to the spatial grid defined for $P(a_t)$, the mass is 'settled' into appropriate bins based on the distance of each bit of mass and the nearest two bins. Mass located in the first and last bin, corresponding to mass that has been captured by the boundary, cannot change locations, and the entries of $M(\theta_a, \delta_t)$ that determines how the mass in these bins moves reflects this. $n=53$ and $\Delta t = 10$ ms for all results presented here.

## Relating *a(t)* to spikes and choices

On a single behavioral trial, the observed spike count of the *n*-th neuron at time *t*, $y_{n,t}$, is a Poisson random variable,

$$P(y_{n,t}|a_t, \theta_n) = (f_{\theta_n}(a_t)) \Delta t^{y_{n,t}} \exp(-f_{\theta_n}(a_t)\Delta t), \tag{9}$$

where $\theta_n$ defines the expected firing rate function *f* for the *n*-th neuron. We choose $f_{\theta_n}$ to be a softplus function, i.e., softplus(x)=log(1+exp(x)). Each neuron has their own parameter $\theta_n$ that relates $f_{\theta_n}$ to $a_t$. $\theta_y = \{\theta_1, \theta_2, ...\theta_N\}$ is the collection of all neural parameters for the population of *N* neurons.

We define $f_{\theta_n}(a_t)$ as

$$f_{\theta_n}(a_t) = softplus\left(\theta_n a_t + \theta_{n,t}^0\right), \tag{10}$$

where $\theta_{n,t}^0$ accounts for the time-varying trial-average (i.e., invariant to *a(t)*) firing rate of the *n*-th neuron. $\theta_{n,t}^0$ is learned prior to fitting the full model, i.e., before learning $\theta_a$ and $\theta_y$. We approximate $\theta_{n,t}^0$ with a set of six Gaussian radial basis functions

$$\theta_{n,t}^0 = \sum_i^6 w_{i,n}^{RBF} N\left(\mu_i, \sigma_{RBF}^2\right) \tag{11}$$

The mean of the functions, $\mu_i$, are spaced uniformly from time 0 to the maximum trial length for each respective neuron. The variance of the functions, $\sigma_{RBF}^2$, is equal to the distance between the function means. We learn $w_{i,n}^{RBF}$ by assuming that $y_{n,t}$ is distributed Poisson with an intensity function $\theta_{n,t}^0$ and maximize the likelihood. In other words, for the *n*-th neuron we define the likelihood of the observed spikes for a trial of duration *T*, $y_n$, assuming a time-varying intensity function $\theta_{n,t}^0$

$$P\left(y_n|\theta_n^0\right) = \prod_{t=1}^{T}\left(\theta_{n,t}^0\Delta t\right)^{y_{n,t}} \exp\left(-\theta_{n,t}^0\Delta t\right), \tag{12}$$

and maximize this likelihood across *K* trials with respect to the parameters $w_{i,n}^{RBF}$.

Although both $\theta_n a(t)$ and $\theta_n^0(t)$ vary in time to define each neuron's expected firing rate, they are uniquely identifiable, because $\theta_n a(t)$ varies from trial to trial depending on the stimulus while $\theta_n^0(t)$ does not. We verified through numerical experimentation and parameter recovery using synthetic data that each process can be identified.

On a single behavioral trial, with a probability $1 - \gamma$ the subject's choice, *d*, is a deterministic function of *a(t)* at the end of the trial (time *T*), (*Brunton et al., 2013*); with probability $\gamma$ the choice is made without considering *a(t)*. $\gamma$ captures 'lapses' in the subject's performance. For choices that depend on *a(t)*, if *a(T)* is greater than a cutoff value *c*, d=1, otherwise d=0. Thus, the probability of the choice, given *a(t)* and $\theta_d$, can be written as:,

$$P(d|a_T, \theta_d) = \left(\frac{\gamma}{2} + (1-\gamma)H(a_T - c)\right)^d \left(\frac{\gamma}{2} + (1-\gamma)(1 - H(a_T - c))\right)^{1-d}, \tag{13}$$

where $H(\cdot)$ is the Heaviside function. We refer to the parameters relating *a(t)* to the likelihood of a subject's choice as $\theta_d = \{c, \gamma\}$.

## Relative binning of clicks and spikes

A minor but key implementation detail concerns defining the start and end times of the temporal bin edges that are used to bin the click inputs and the spikes trains. Through numerical experimentation, we identified that our numerical procedure produces a systematic error in estimating the model

parameters when the temporal bins for the clicks are aligned with the temporal bins for the spikes. To circumvent this issue, we offset the bins for the spikes by $\Delta t/2$, so that the bin edges for spikes at time $t$ surround the forward bin edge of the clicks by $\pm\Delta t/2$. This procedure is similar to the central difference formulation of a finite difference approximation to a differential equation.

## Inferring model parameters with maximum likelihood

We refer to the set of all parameters for models fit to spikes and choices as $\Theta = \{\theta_a, \theta_y, \theta_d\}$. Given the Markov dynamics described above, the relationship between $a(t)$ and the observed data, and the model parameters, we can write out the likelihood of the spike train data $Y$ from $N$ neurons for $T$ time bins, the behavioral choice $d$, and the latent variable $a$ for $T$ time bins as:

$$P\left(a, Y, d|\Theta\right) = P\left(a_0|\theta_a\right) \prod_{t=1}^{T} P\left(a_t|a_{t-1}, \theta_a, \delta_t\right) \prod_{n=1}^{N} P\left(y_{n,t}|a_t, \theta_n\right) P\left(d|a_T, \theta_d\right) \tag{14}$$

We compute the likelihood of the data by integrating over $a$

$$P\left(Y, d|\Theta\right) = \sum_a P\left(a, Y, d|\Theta\right) \tag{15}$$

Because of the way in which we compute $P(a_t|a_{t-1}, \theta_a, \delta_t)$ (see above) computing the log-likelihood of the data can be done with a single forward pass over the data using the 'forward-backward' algorithm method for hidden Markov models (**Bishop, 2006**). We maximize the sum over $K$ behavioral trials of the logarithm of this quantity with respect to $\hat{\Theta}$ via gradient ascent. To compute the gradient of $\sum_k^K log\, P(Y_k, d_k|\Theta)$ with respect to $\Theta$, we use a standard automatic differentiation package (**Revels et al., 2016**). We refer to the set of parameters that maximizes the likelihood as $\hat{\Theta}$.

We note that all $K$ trials for many of the models we fit were not recorded on the same behavioral session, and therefore, all $N$ neurons are not recorded for every trial. For example, neurons 1–3 might be recorded on trials 1–500, while neurons 4–6 might be recorded on trials 501–1000. Although our notation does not reflect this in order to keep the notation simple, only neurons recorded on a trial contribute to the likelihood on that trial.

## Bounded optimization

Several model parameters are only defined within a restricted domain; for example, all variances parameters, such as $\sigma_a^2$, are only defined on the positive real axis. Alternatively, other parameters, although defined on a more expansive domain, have values that correspond to models that are not very likely; for example, although $B$ is defined on the positive real axis, values much greater than 40 are not likely to be exhibited in the data, given the specifics of the stimulus, where greater than 40 clicks were rare. For these reasons, we define the following domain over which parameter optimization was performed:

- $1e^{-3} \leq \sigma_a^2 \leq 100$
- $8 \leq B \leq 40$
- $-5 \leq \lambda \leq 5$
- $1e^{-3} \leq \sigma_a^2 \leq 400$
- $1e^{-3} \leq \sigma_s^2 \leq 10$
- $1e^{-3} \leq \phi \leq 1.2$
- $5e^{-3} \leq \tau_\phi \leq 1$
- $-10 \leq c \leq 10$
- $0 \leq \gamma \leq 1$
- $-10 \leq \theta_n \leq 10 \forall n$

The occurrence of parameters hitting the bound can be seen in *Figure 3*, *Figure 3—figure supplement 4*. The most common boundary hitting situation was a variance parameter ($\sigma_i$, $\sigma_a$, $\sigma_s$) hitting the lower boundary of zero, which means that the model did not support noise of that kind in the model fit. $\sigma_i$ and $\sigma_a$ were found to do this for the choice-only model, consistent with the results of Brunton et al. The other bound that was frequently hit was the upper bound for the accumulation bound parameter $B$, a result also consistent with the results of Brunton et al. The log-likelihood surface as $B$ grows very large becomes very flat, because it becomes increasingly unlikely that probability mass $P(a(t))$

crosses the boundary. Thus, the model fits do not change appreciably if this optimization boundary is relaxed.

## Confidence intervals for maximum likelihood parameters

To compute confidence bounds of estimated parameters (as in *Figure 3* and *Figure 1—figure supplement 1*, *Figure 4—figure supplement 1*, *Figure 3—figure supplement 3*, *Figure 3—figure supplement 4*), we use the Laplace approximation to the log-likelihood. Using automatic differentiation, we compute the Hessian (the matrix of second derivatives) of the log-likelihood at the maximum likelihood parameters. The diagonal entries of the Hessian's inverse quantify the sharpness of the curvature of the log-likelihood surface, and therefore the uncertainty of the estimate of each parameter. We define the confidence bound as ±2 times the square root of each diagonal entry; approximating the log-likelihood surface as Gaussian, this describes the range of parameters that would fall within approximately 95% of the log-likelihood volume.

For some sets of maximum likelihood parameters, further consideration was necessary. In cases where confidence bounds extend beyond an optimization bound that corresponds to a strict boundary on the domain of a parameter (e.g., variance parameters being strictly positive), we truncate these intervals at the bound. In some cases, we found that Hessian was not positive semi-definite, a necessary condition to invert it. This most often occurred when a maximum likelihood parameter encroached upon a strict parameter boundary (e.g., variance parameters being strictly positive). We dealt with these scenarios in two ways. In some cases, numerical line search along any Hessian eigenvector with negative eigenvalue confirmed the convexity of the log-likelihood was local whereas more globally the log-likelihood was concave. In light of this, we numerically computed the global concavity of the log-likelihood with a numerical line search and approximated this curve with a quadratic function. We replaced the negative eigenvalue of the Hessian with two times the coefficient of this quadratic approximation (the multiplier two is used because the Hessian is two times the second-order approximation of the log-likelihood via Taylor series approximation, where the second-order term contains a 1/2 prefactor). In other cases, computing the Hessian in a transformed space (e.g., log space) where troublesome parameters were free to take on any value rectified the non-concavity (*Yartsev et al., 2018*). After computing confidence intervals in the transformed space, we mapped these values back into the standard space by the inverse transform.

## Data selection

Details regarding behavioral data collection and neural recordings and spike sorting can be found in *Hanks et al., 2015* and *Yartsev et al., 2018*. To select which neurons were used, a firing rate for each neuron was computed by summing spikes over the duration of the stimulus period and dividing this by the length of the stimulus period. A two-sided t-test was applied, comparing the firing rate distribution on trials when the animal chose left and when the animal chose right. Neurons with a p-value less than 0.01 were included for analysis.

## Data grouping

We grouped together rats that had neural recordings from the same brain region (five FOF rats, three PPC rats, three ADS rats; see *Table 1* for information about the data) to improve our estimation of the model parameters for each region. For the PPC and ADS recordings, the majority of recorded neurons came from a single rat (*Table 1*). Although individual FOF rats had enough neurons to support fitting each rat alone, the maximum likelihood parameters for FOF rats fit individually were qualitatively similar (*Figure 3—figure supplement 3*).

## Response latency

Previous analyses have identified a response latency between the stimulus and the neural responses, and that this latency can be different in different brain regions (*Hanks et al., 2015*). To account for this, we shifted the time of the neural responses relative to the clicks based on these prior results. FOF and ADS responses had a latency of 60 ms, while PPC responses had a latency of 120 ms.

## Specifics of data selection for each analyses

Our reports of the maximum likelihood parameters for each model are for models fit to the entire dataset. Each model was also fit using cross-validation (i.e., training on a subset of the data, while reserving data for testing) but the maximum likelihood parameters did not qualitatively change from those identified using the entire dataset, and the log-likelihood computed on test data using parameters identified with training data did not differ appreciably from the log-likelihood computed on those same trials using parameters identified with the entire dataset (*Figure 3—figure supplement 4*). This consistency is likely due to the modest number of model parameters.

When we compute various quantities related to the data, such as PSTHs, cross-correlation functions, and psychometric functions, we likewise use the entire dataset. We did not find that we could accurately estimate the PSTH when only using a small subset of the data (i.e., test data) due to the fact that our task lacks repeated stimulus conditions. Additionally, when we simulate data from a fit model (e.g., *Figure 2A*), we used the maximum likelihood parameters derived from model fits to the entire dataset, and used the stimuli of the entire dataset to generate these data. Again, because the maximum likelihood parameters did not qualitatively change when the model was fit to a subset of the data, we found it easier to focus our analyses on a single model. The above statements apply to analyses in the following figures: *Figure 2*, *Figure 3*, *Figure 4C, D*, *Figure 5*, *Figure 6B–F*, *Figure 3—figure supplement 2*, *Figure 4—figure supplement 1*, *Figure 3—figure supplement 3*, *Figure 6—figure supplement 2*.

When comparing performance across models, cross-validation is necessary, and we did so in those cases (e.g. *Figure 4B*, *Figure 6A*, *Figure 4—figure supplement 2*, *Figure 6—figure supplement 1*, *Figure 3—figure supplement 4B*). In these cases, we performed fivefold cross-validation by dividing the dataset into a training set that consisted of 80% of the data and a test set that consisted of 20% of the data. We fit each model using the training data of each fold, and computed the test log-likelihood using the test data and the parameters derived from the training data. Test performance was averaged across the five folds. Again, we stress that the test performance on cross-validated data did not appreciably differ from that computed using a model trained to the entire dataset (*Figure 3—figure supplement 4*). We note, however, that even in cases when we performed cross-validation, we still computed an approximation to each neuron's trial-averaged firing rates, $\theta_{n,t}^0$, using all available data, prior to fitting the full model.

Because most of our models were fit simultaneously to data from multiple experimental sessions (in which different neurons are recorded), to perform cross-validation, we randomly divided trials within each session into a train and test set, and trained and tested the model collectively on those groups of trials. Testing the model in this way will determine parameter robustness across all sessions (for model parameters that are shared across all sessions) and individual parameter robustness within a session (for parameters that are specific to an individual session). This procedure also worked for the 'independent-noise model', for which model parameters were shared across all sessions, but individual neuron parameters were session specific.

## Other fit models

### Independent-noise accumulator models

We refer to the set of all parameters for the model with independent accumulator noise per neuron as $\Theta_{ind}$. The likelihood of the spike train data from the $n$-th neuron $Y_n$ for $T$ time bins is

$$P\left(Y_n | \Theta_{ind}\right) = \sum_{a_n} P\left(a_{0,n} | \theta_a\right) \prod_{t=1}^{T} P\left(a_{n,t} | a_{n,t-1}, \theta_a, \delta_t\right) P\left(y_{n,t} | a_{n,t}, \theta_n\right) \tag{16}$$

The joint likelihood for the spike train data from all neurons is the product of the likelihood for each neuron: $P(Y|\Theta_{ind}) = \prod_{n=1}^{N} P(Y_n|\Theta_{ind})$. Our primary interest in this analysis was capturing the neural responses, so we considered a simple model of choice for this model: on each trial, choice is determined by randomly selecting one of the accumulators. The likelihood of the choice $d$ under such a model is the average of the $n$ accumulators at time $T$:

$$P\left(d | \Theta_{ind}\right) = \frac{1}{N} \sum_{n=1}^{N} P\left(d | a_{n,T}, \theta_d\right) \tag{17}$$

The full likelihood is the product of these terms: $P\left(Y, d|\Theta_{ind}\right) = P\left(d|\Theta_{ind}\right) P\left(Y|\Theta_{ind}\right)$.

## Choice-only model

We refer to the set of all parameters for the model fit to choices only as $\Theta_d = \{\theta_a, \theta_d\}$. The likelihood of the behavioral choice $d$ is

$$P\left(d|\Theta_d\right) = \sum_a P\left(a_0|\theta_a\right) \prod_{t=1}^{T} P\left(a_t|a_{t-1}, \theta_a, \delta_t\right) P\left(d|a_T, \theta_d\right) \qquad (18)$$

## Bernoulli GLM

To benchmark our method's ability to predict the animal's choice, we considered a basic logistic regression model (i.e., Bernoulli GLM) that included stimulus information and neural activity (e.g., **Figure 6A** and **Figure 6—figure supplement 1**). For each trial, we computed the total number of spikes each neuron produced during the specified temporal window and the final cumulative click difference, and used them as regressors in a standard Bernoulli GLM to predict the animal's choice. A constant bias was also included, as well as a single lapse parameter that scaled the minimum and maximum values of the logistic inverse link function. Cross-validation was performed on this model as described above.

## Null choice model

In **Figure 6A**, we assess how well each of our fitted models can predict choice. We compare all models against a baseline model where each choice is a Bernoulli random variable with probability of making a right choice equal to the empirical fraction of choices made to the right.

## Null joint model

To compare the improvement of the joint model in absolute terms (i.e., when not comparing two fitted models), we compute a null model of the spiking activity and choices (**Figure 3—figure supplement 4B**). The null likelihood of the choice data is as described above. The null likelihood of the spike train data assumes that the time-varying expected firing rate of each neuron is equal to its estimated time-varying trial-average firing rate, i.e., $f_{\theta_n}\left(t\right) = \theta_{n,t}^0$.

The improved performance (i.e., cross-validated log-likelihood) of our joint model over the null model shown in **Figure 3—figure supplement 4** further confirms that $\theta_n a\left(t\right)$ and $\theta_n^0\left(t\right)$ are uniquely identifiable, and that they are not redundant (i.e., the joint model is not overparameterized).

## Poisson GLM

To validate the maximum likelihood parameters derived from the joint model, we fit a variant of a Poisson GLM to the spiking responses (**Figure 3—figure supplement 2**). As a regressor, we used the adapted, exponentially filtered click inputs,

$$da = \lambda a dt + dt\Delta\left(t\right), \qquad (19)$$

where $\Delta\left(t\right)$ is defined as above. The expected firing rate of each neuron is defined as in the full model, by **Equation 10**. For the bounded Poisson GLM model, the dynamics of $a(t)$ follow **Equation 19**, except that if $a(t)$ crosses $B$, $a(t)$ stops evolving (i.e. $da = 0$ if $a(t)>B$). The parameters $\lambda$, $B$, $\phi$, $\tau_\phi$, and $\theta_y$ that maximize the likelihood of the spike data were learned using gradient ascent. The null model described in **Figure 3—figure supplement 2** is the null joint model, described above.

## Negative binomial

In **Figure 3—figure supplement 1**, we compare a Poisson observation model to a negative binomial model. To do this, we model the spikes as:

$$P\left(y_{n,t}|a_t, \theta_n\right) = NB\left(\theta_n^{NB}, \frac{\theta_n^{NB}}{f_{\theta_n}\left(a_t\right)\Delta t + \theta_n^{NB}}\right) \qquad (20)$$

where $NB\left(\cdot,\cdot\right)$ is the negative binomial distribution, and $\theta_n^{NB}$ controls the variance of the distribution for each neuron and can take values between 0 and positive infinity. When $\theta_n^{NB}$ becomes large the negative binomial distribution approaches the Poisson distribution. $\theta_n^{NB}$ was fit for each neuron using gradient ascent, as described above.

## Quantifying model fit

### Computing PSTHs and cross-correlation functions on empirical data

We computed a 'single-trial' firing rate for each neuron by convolving its binned spikes with a Gaussian kernel of standard deviation 50 ms. We call this single-trial rate $r_{t,k,n}$ for the $n$-th neuron on the $k$-th trial at time $t$. We divide all the trials into two equally sized groups based on the cumulative click difference at the end of the trial and average $r_{t,k,n}$ based on these groupings. Because trials are not of equal duration, at time $t$ we use whichever trials have data at that time. We refer to this average as $\bar{r}_{c,n,t}$ where the index $c$ runs from 1 to 2.

We used the empirical binned spikes counts to compute cross-correlation functions. Raw cross-correlation functions were normalized by the (across time) mean firing rates of the two neurons being used so they provided a measure of excess spike rate. The equation for the raw cross-correlation function was

$$R_{m,n}\left(\tau\right) = \frac{1}{m_m}\left(\frac{1}{K}\sum_k^K\frac{1}{N_k\left(\tau\right)}\sum_t^T\frac{y_{n,k,t}}{\Delta t}\frac{y_{m,k,t-\tau}}{\Delta t}\right) - m_n, \tag{21}$$

where $t$ is over all bins for the $k$-th trial, $y_{n,k,t}$ and $y_{m,k,t-\tau}$ are the binned spike train of neuron $n$ and $m$ at time $t$ and $t-\tau$ respectively, and $N_k\left(\tau\right)$ is the number of bins such that both $y_{n,k,t}$ and $y_{m,k,t-\tau}$ are valid. $m_n$ and $m_m$ are the mean firing rates of the $n$-th and $m$th neuron respectively, computed by taking the average spike count across all times.

To compute the shuffled-corrected cross-correlation, we computed the cross-correlation of the expected firing rate of each neuron provided by the PSTH, i.e., $\bar{r}_{n,c,t}$,

$$R_{m,n}^{PSTH}\left(\tau\right) = \frac{1}{m_m}\left(\frac{1}{C}\sum_c^C\frac{1}{N_c\left(\tau\right)}\sum_t^T\bar{r}_{n,c,t}\bar{r}_{m,c,t-\tau}\right) - m_n, \tag{22}$$

where $C$=2 is the number of conditions used to define the PSTH, $N_c\left(\tau\right)$ is defined similarly as above, and $m_n$ and $m_m$ are as defined above. The shuffle-corrected cross-correlation is the raw cross-correlation minus the cross-correlation of the expected firing rate: $R_{m,n}\left(\tau\right) - R_{m,n}^{PSTH}\left(\tau\right)$.

### Computing PSTHs and cross-correlation functions on synthetic data

We generated synthetic data from a model by using the maximum likelihood parameters to generate the expected firing rate of each neuron on each trial, i.e., $f_{t,k,n}$. We averaged this expected rate for each neuron on each trial over 20 different realizations of the latent noise to reduce variation due to the latent process. We then grouped and averaged these average expected rates, as described above, to generate a synthetic PSTH, which we denote by $\bar{f}_{n,c,t}$, as used in *Figure 2* and *Figure 1—figure supplement 1*.

We used the synthetic expected firing rate, $f_{t,k,n}$, to compute cross-correlation function for synthetic data,

$$R_{m,n}^{syn}\left(\tau\right) = \frac{1}{m_m}\left(\frac{1}{K}\sum_k^K\frac{1}{N_k\left(\tau\right)}\sum_t^T f_{n,k,t}f_{m,k,t-\tau}\right) - m_n, \tag{23}$$

where $K$, $N_k\left(\tau\right)$, $m_n$, and $m_m$ are as defined above. The shuffle-corrected cross-correlation function of synthetic data is the raw cross-correlation function minus the cross-correlation function of the expected synthetic firing rate provided by the synthetic PSTH, $\bar{f}_{n,c,t}$.

### Goodness-of-fit metrics

To compare empirical and synthetic PSTHs, we computed the coefficient of determination. Because fewer and fewer trials were included in computing the PSTH at large time values (because trials of

great length were rare), we included PSTH values 200 ms before the stimulus onset up until 500 ms after stimulus onset in this calculation. Based on the definitions of the empirical and synthetic PSTHs, the coefficient of determination is defined as:

$$R_n^2 = 1 - \frac{\sum_c^C \sum_t^T \left( \bar{r}_{n,c,t} - f_{n,c,t} \right)^2}{\sum_c^C \sum_t^T \left( \bar{r}_{n,c,t} - <\bar{r}_{n,c,t}>_{c,t} \right)^2}, \tag{24}$$

where $<\bar{r}_{n,c,t}>_{ct}$ is the mean of $\bar{r}_{n,c,t}$ over trial groupings and times. Pearson correlation ($r$) was used to compare empirical and synthetic cross-correlation functions. When computing $r$ we considered values of $\tau$ between –800 and 800 ms.

## Psychometric functions

We used a Bernoulli GLM (i.e., logistic regression) to compute psychometric functions for empirical and synthetic data. We generated synthetic data from a model by using the maximum likelihood parameters to generate the probability of a choice, and sampled the choice from a Bernoulli distribution. For the Bernoulli GLM, for each trial, we computed the final click difference and used it as a regressor to predict the animal's choice. A constant bias was also included, as well as a single lapse parameter that scaled the minimum and maximum values of the logistic inverse link function. $R^2$ values comparing empirical and synthetic psychometric functions were defined as above, but using the psychometric functions whose domain was from the minimum final cumulative click difference to the maximum final cumulative click difference.

## Choice decoding

We used two metrics to determine how well choice could be decoded from various models: choice-prediction accuracy and test log-likelihood. Test likelihood was reported in bits per trial, i.e.,

$$\Delta LL = \frac{LL_{model} - LL_{null}}{log_2 (K)} \tag{25}$$

where $K$ is the number of trials in the test set and $LL_{null}$ is the appropriate null model, as described above, or a second model with which to test against. Fivefold cross-validation was performed, as described above. Accuracy was determined, depending on the model, by computing the probability that the model predicted a right choice, given all available data (i.e. inputs and spikes in a model that includes spikes). If the model had a greater than 0.5 probability of choosing right, we considered that a prediction of a rightward choice. Accuracy is the fraction of correct choice predictions.

## Identifying putative changes of mind

Based on a recent study (*Peixoto et al., 2021*) we defined putative changes in mind in the following way. For each model and each trial, we computed the posterior distribution of $a(t)$ given all available data except for the choice. In the case of the choice-only model, this means using only the stimulus, and is equivalent to the forward pass of the model. In the case of the joint model, this is equivalent to the posterior distribution of $a(t)$ given the spikes on that trial. We computed the expected value of the posterior distribution for each trial and identified moments when it crossed the decision threshold as determined for each model (i.e., the $c$ parameter of the choice likelihood). We required that the expected value remain on one side of the threshold for 50 ms, remain on the other side following the crossing for 50 ms, and achieve an absolute magnitude greater than or equal to 2 at some point during that 100 ms window.

To relate putative change of mind events to the animal's behavior, we performed linear regression between the time of the event relative to the end of the stimulus (i.e., how close to a decision the event occurred) and a measure of the animal's reaction time. In this task, the animal is required to fixate in the center poke for the duration of the stimulus, so it does not exhibit a true reaction time in the standard sense of the term. However, following the end of the stimulus, it does take the animal time to withdraw from the center port to make its choice (see *Figure 1A*, bottom, upper two lines). We refer to the difference between the end of the stimulus and when the animal withdrew from the center port as the animal's reaction time, which we used in our analysis.

## Estimating dimension

To estimate the effective dimension of groups of simultaneously recorded neurons, we computed the 'participation ratio' (*Litwin-Kumar et al., 2017*). Single-trial firing rates were computed by convolving the spike trains with a Gaussian kernel (std = 50 ms), and the covariance matrix of these rates was computed. The participation ratio is

$$\frac{\left(\sum_n^N \lambda_n\right)^2}{\sum_n^N \left(\lambda_n\right)^2}, \tag{26}$$

where $\lambda$ are the eigenvalues of the covariance matrix. If the firing rates are independent, the eigenvalues will all be equal and the participation ratio will equal the number of neurons. If the firing rates are correlated such that some eigenvalues are small (or perhaps even zero) the participation ratio will be less than the number of neurons.

## Acknowledgements

This work was supported by Simons Collaboration on the Global Brain (SCGB AWD543027 and AWD542593), and NIH-NINDS BRAIN Initiative Award (5U19NS104648-02). We thank Michael Yartsev, Tim Hanks, and Charles Kopec for providing the neural and behavioral data analyzed here. We thank members of the Brody and Pillow labs for comments on this work, especially Thomas Luo, Tim Kim, Diksha Gupta, and Tyler Boyd-Meredith. BD would like to thank Carol Mason, Paul DiMaggio, Betsy Levy Paluck, and Nathan Paluck for their support while this study was conducted.

## Additional information

### Competing interests

Carlos D Brody: Reviewing editor, eLife. The other authors declare that no competing interests exist.

### Funding

| Funder | Grant reference number | Author |
|---|---|---|
| Simons Foundation | SCGB AWD543027 | Carlos D Brody Jonathan W Pillow |
| Simons Foundation | SCGB AWD542593 | Carlos D Brody |
| National Institute of Neurological Disorders and Stroke | BRAIN Initiative Award 5U19NS104648-02 | Jonathan W Pillow Carlos D Brody |
| National Institute of Biomedical Imaging and Bioengineering | R01 EB026946 | Jonathan W Pillow |
| National Institute of Biomedical Imaging & Bioengineering | 9RF1DA065404 - 04 | Jonathan W Pillow |

The funders had no role in study design, data collection and interpretation, or the decision to submit the work for publication.

### Author contributions

Brian DePasquale, Conceptualization, Data curation, Software, Formal analysis, Investigation, Visualization, Writing – original draft, Writing – review and editing; Carlos D Brody, Jonathan W Pillow, Conceptualization, Supervision, Funding acquisition, Project administration, Writing – review and editing

### Author ORCIDs

Brian DePasquale ⓘ https://orcid.org/0000-0002-3830-3184
Carlos D Brody ⓘ https://orcid.org/0000-0002-4201-561X

Jonathan W Pillow [ID] https://orcid.org/0000-0002-3638-8831

**Decision letter and Author response**
Decision letter https://doi.org/10.7554/eLife.84955.sa1
Author response https://doi.org/10.7554/eLife.84955.sa2

## Additional files

### Supplementary files
MDAR checklist

### Data availability
All code was written in the Julia programming language. The core codebase for fitting the models described in this manuscript can be found at https://github.com/Brody-Lab/PulseInputDDM.jl (copy archived at *Brody-Lab, 2024a*). All data analyzed during this study and original analysis computer code has been deposited at https://github.com/Brody-Lab/DePasquale-eLife-2024 (copy archived at *Brody-Lab, 2024b*).

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
