## [Editor Report]

This valuable paper presents findings showing that different brain regions were best described by a distinct accumulation model, which all differed from the model that best described the rat's choices. These findings are solid because the authors present a very strong methodological approach. This work will be of interest to a wide neuroscientific audience.

---

## [Decision Letter]

**Decision letter after peer review:**

Thank you for submitting your article "Neural population dynamics underlying evidence accumulation in multiple rat brain regions" for consideration by *eLife*. Your article has been reviewed by 2 peer reviewers, and the evaluation has been overseen by a Reviewing Editor and Michael Frank as the Senior Editor. The following individual involved in the review of your submission has agreed to reveal their identity: Brandon Turner (Reviewer #1).

Essential revisions (for the authors):

1. I found some of the decisions about the model development to be somewhat unusual, and I wondered whether the results hinged on these assumptions. To be clear, I am fine with unusual, but I was not clear how the authors came to the decisions they made. I think it would be better to describe why some decisions were made. For example, Equation 1 uses both the sum and difference of the evidence directly in the accumulation dynamics, but I wonder why the authors used this expression and not the standard DDM. It was also unclear why leakage was added in this way (where λ can be negative) for a model that uses a two-boundary setup (i.e., a non-racing accumulator structure). Other unusual things were the step function combined with a contaminant process (lapse) to relate to a probability of choice and a softplus function for the neural activity (Equations 2 and 3). Because these were choices I was unfamiliar with, I wondered where they were from and whether their incorporation had any impact on the results.

2. I also felt that the paper was lacking some connection to other joint modeling efforts that use trial-by-trial parameters to link neural and behavioral data. These are not quite the same as the authors' approach, but it could be good to link to those many lines of research to leave some 'breadcrumbs' for other researchers who are interested in modeling brain-behavior links.

3. Why are there two absorbing bounds, one for a(t) and one for the choice criterion? Evidence accumulation models typically impose an absorbing bound for (an analogue of) a(t), and assume a choice is made when that bound is reached. Can the authors clarify the purpose of deviating from this assumption?

4. Page 13: Reference to Figure 4A might mean 4B.

5. Equations 10-11: I'm wondering to what extent there might be a collinearity issue here. These models allow firing rates to vary over time as a function of two mechanisms: \theta_{n}a_t, which is time-varying because a varies over time; and \theta_{n,t}^{0}, which is time-varying by itself through equation 11. I was wondering: If firing rates indeed covary with a, then doesn't the model have two options to model this: both via \theta_{n} and via \theta_{n,t}^0?

This point is especially relevant for the section on the independent noise accumulator models, where \theta_{y} is fit for every neuron individually, and as such, these parameters are informed by relatively little information which might increase the uncertainty on these parameters. Are the results shown in Figure 4B not potentially an overfitting issue? A related question here pertains to the cross-validation procedure: How exactly are the data partitioned? If the parameters fit on the individual neuron level, then the split between train and test data should only split trials of the same neurons (under the independent noise model, \theta_{1,T} cannot be expected to predict \theta_{2,T}, as these are different neurons, I would think? Or am I missing something?).

6. Page 39: were the bounds of optimization ever reached?

7. Null joint model: This is related to the point above, but I'm not sure if Figure S9B (referred to on page 42) actually shows the results of this model. I was indeed wondering how well this model cross-validates, in light of the potential collinearity issue raised in point 3 above.

8. Figure S9A: I don't see '+'-symbols.

9. Typos.

Page 21, Figure S1C title: "Example recovered parameters" (missing e in parameters).

Page 36: "The transition matrix M(θa, δt) it is determined using methods established in Brunton (2013)". "it" should probably be removed.

10. Code availability:

The authors state they will make the code publicly accessible upon publication. It would be useful to include a persistent link (to e.g. osf or github) in the manuscript to facilitate finding the manuscript after it has been published. The code itself could still remain under embargo as long as the review lasts, should the authors prefer this.

---

## [Author Response]

Essential revisions (for the authors):1. I found some of the decisions about the model development to be somewhat unusual, and I wondered whether the results hinged on these assumptions. To be clear, I am fine with unusual, but I was not clear how the authors came to the decisions they made. I think it would be better to describe why some decisions were made. For example, Equation 1 uses both the sum and difference of the evidence directly in the accumulation dynamics, but I wonder why the authors used this expression and not the standard DDM. It was also unclear why leakage was added in this way (where λ can be negative) for a model that uses a two-boundary setup (i.e., a non-racing accumulator structure). Other unusual things were the step function combined with a contaminant process (lapse) to relate to a probability of choice and a softplus function for the neural activity (Equations 2 and 3). Because these were choices I was unfamiliar with, I wondered where they were from and whether their incorporation had any impact on the results.

Our apologies for the lack of clarity. In equation (1), we should have clarified how our choices are closer to the standard DDM than they might appear. Our motivation for selecting the parameters of the latent accumulator as we did was the findings presented in Brunton et al. 2013. In this study, the authors fit an identical model to the choices of rats and humans performing the same pulse-based evidence accumulation task and found that the model was sufficiently flexible to characterize the variety of behavioral strategies they employed (see Figure 2E-J of Brunton et al. 2013). For this reason, we felt it also presented an appropriate and flexible starting point for considering how well neural activity reflected accumulation dynamics. Fitting this model jointly to neural activity and choices, or choices alone, allowed us to understand how incorporating neural responses into the model compared to the existing literature related to this model and rat behavior. We now state this motivation more explicitly:

“The essence of our model is to describe a DDM-based accumulation process driven by sensory stimuli following Brunton et al., 2013 and relate the latent accumulation process to both neural responses and the rat’s choice. Previous results have shown that this model is sufficiently flexible to accommodate the various behavioral strategies rats exhibit while performing this task (Brunton et al., 2013).”

Regarding why the sum and difference were used, the difference reflects the moment to moment evidence in favor of a choice (left vs. right) while the sum is necessary to ensure that the magnitude of the sensory noise scales with the number of sensory pulses experienced at each moment in time. We now explain this more carefully:

“The final term, σsΣ(t)ηdt, introduces noise into *a(t)* that is proportional to the total number of clicks that occur at a given moment. The sum of clicks Σ(t) is included so that the magnitude of the noise increases depending on the number of sensory clicks experienced at time *t*.”

Regarding the leak term, again following Brunton, we wanted to allow the model the flexibility to fit the various behavioral strategies that rats were found to exhibit, including leaky accumulation (negative λ) and impulsive decision-making (positive λ). This proved to be crucial, as we found that the FOF was best described by a positive λ, an intriguing feature of the data that would have been otherwise overlooked. We have modified the text to indicate this:

“Previous results have shown that rats exhibit a range of accumulation strategies spanning these values of λ (Brunton et al., 2013).”

Regarding the step function with lapses to model choice, in our experience, this choice is standard: a step function being the deterministic discriminant to map a continuous variable (accumulated evidence) to a binary choice, and lapses being the standard method for modeling choices that do not depend on the stimulus. An alternative way to view this choice is as an infinitely steep logistic function. These choices were based on the approach of Brunton et al. We have modified the text to explain our choice:

“Previous work has found that parameterizing choice this way creates a model that is sufficiently flexible to describe animals’ choice (Brunton et al., 2013) while remaining as simple as possible.”

The softplus function for determining neural activity is a standard choice in our experience. The softplus function is often chosen when relating a variable of interest to the spike count of a neuron because it is the simplest rectified function that is differentiable (it is a differentiable approximation to a rectified-linear function). The work that is most related to our is Latimer et al. 2015 which uses a softplus. We have modified the text to explain our choice:

“The softplus function (smooth rectified linear function) was used to ensure the expected firing rate was positive, and was selected because it is the simplest function to achieve this goal, and also based on prior success in similar studies (e.g., Latimer et al. 2015).”

2. I also felt that the paper was lacking some connection to other joint modeling efforts that use trial-by-trial parameters to link neural and behavioral data. These are not quite the same as the authors' approach, but it could be good to link to those many lines of research to leave some 'breadcrumbs' for other researchers who are interested in modeling brain-behavior links.

We thank the reviewers for catching this and we apologize for failing to link our work to important existing results. We have added a paragraph to the Discussion section highlighting these works, including several new references, and how they relate to our results:

“Our approach extends and complements existing approaches that construct formal mathematical models of decision making which combine both behavioral data and neural data. These models leverage both neural and behavioral observations to jointly infer decision making parameters, as we’ve done here (see Turner et al., 2019 for a comprehensive overview). However, the majority of these approaches have tended to emerge from the field of cognitive neuroscience, and as such, have predominantly focused on models for application to neural data acquired by other methods, such as EEG, fMRI, etc. (e.g., Turner et al. 2015; but also see Frank et al., 2015). Our approach adds to these efforts by offering a method that can combine fine timescale single unit recordings with behavioral measurements specifically during pulse-based evidence accumulation tasks, thereby offering a moment-by-moment picture into the latent dynamics that underlies cognition. Continued development of joint models such as our and existing approaches in the field of cognitive neuroscience are critical to quantitatively understand the latent processes underlying cognition.”

New References:

Turner, B.M., Forstmann, B.U., Steyvers, M, 2019. Joint Models of Neural and Behavioral Data, Springer International Publishing.

Turner BM, van Maanen L, Forstmann BU. Informing cognitive abstractions through neuroimaging: the neural drift diffusion model. Psychol Rev. 2015 Apr;122(2):312-336. doi: 10.1037/a0038894. PMID: 25844875.

Frank MJ, Gagne C, Nyhus E, Masters S, Wiecki TV, Cavanagh JF, Badre D. 2015. fMRI and EEG predictors of dynamic decision parameters during human reinforcement learning. J Neurosci. 2015 Jan 14;35(2):485-94. doi: 10.1523/JNEUROSCI.2036-14.2015. PMID: 25589744; PMCID: PMC4293405.

3. Why are there two absorbing bounds, one for a(t) and one for the choice criterion? Evidence accumulation models typically impose an absorbing bound for (an analogue of) a(t), and assume a choice is made when that bound is reached. Can the authors clarify the purpose of deviating from this assumption?

We apologize for the confusion. There is only a single symmetric absorbing boundary that dictates the dynamics of *a(t)*. It has a magnitude of *B* and there are boundaries at +*B* and -*B*. A choice commitment is made when the bound is reached. We have modified the text to explain this more clearly:

“As described above, when *a(t)* crosses the decision bound *B* a choice commitment is made, either to the left or the right, and no further evidence accumulation occurs.”

4. Page 13: Reference to Figure 4A might mean 4B.

Thank you, we have made this correction.

5. Equations 10-11: I'm wondering to what extent there might be a collinearity issue here. These models allow firing rates to vary over time as a function of two mechanisms: \theta_{n}a_t, which is time-varying because a varies over time; and \theta_{n,t}^{0}, which is time-varying by itself through equation 11. I was wondering: If firing rates indeed covary with a, then doesn't the model have two options to model this: both via \theta_{n} and via \theta_{n,t}^0?

Thank you for bringing up this subtle issue. θna(t) and θ0n(t) are not collinear, and are both identifiable because the trajectory of a(t) varies from trial to trial (via its dependence on the stimulus) whereas θ0n(t) does not. The θ0n(t) term captures the component of time-varying firing rate that depends on the arrival times of the clicks in any given trial. (For example, a trial with no clicks present will have the θ0n(t) term but the θna(t) term will be zero). Because of this θna(t) will capture changes in firing rate specific to an individual trials accumulated evidence while θ0n(t) will only capture changes in firing rate that occur in time on *all* trials, regardless of the stimulus.

To reassure the reviewers, we conducted two numerical experiments. First, we simulated data from a simplified model in which *a(t)* was sampled from a Wiener process, i.e. a Gaussian random walk, for *T* timepoints and across a set of *K* trials (see Author response image 2). This is a logical simplification of our more complicated model because a random walk assumes perfect integration of a Gaussian random variable as an input. In other words, we are making two simplifications from our more complicated model: integration is perfect and the inputs are Gaussian not Poisson. We modeled the trial-dependent term of the expected firing rate (i.e., θna(t)) with θn=1 and the trial-independent term as a linear function with one parameter: i.e. θ0n(t)=t. These simplifications are not likely to alter the core finding. We computed the Pearson correlation between *a(t)* and *t* for all *K* trials: it was roughly 0, i.e. uncorrelated. The pseudocode (Author response image 1) recapitulates the simplified model:

**Author response image 1. sa2fig1:** 

Author response image 2 is an image of the simulated data, in which the computed correlation was -0.003:

Thus in this simplified model we found that these two temporal components were not conlinear. Again the central reason for this is that *a(t)* varies across trials while θ0n(t) does not.The second experiment was to examine if generative parameters could be uniquely recovered. To do this, we assumed an expected firing rate of a neuron as we did in our model, but assumed the simplified form for the underlying processes, i.e., *a(t)* was as a Gaussian random walk and the expected firing rate was determined by λn=θna(t)+ θ0n(t)=θna(t)+ θ0nt. We found that when we simulated data with known values of θn and  θ0n that they could be reliably recovered.

To compactly summarize these finding in the manuscript, we have added the following text to the Methods:

“Although both θna(t) and θ0n(t) vary in time to define each neuron’s expected firing rate, they are uniquely identifiable, because θna(t) varies from trial to trial depending on the stimulus while θ0n(t) does not. We verified through numerical experimentation and parameter recovery using synthetic data that each process can be identified.”

This point is especially relevant for the section on the independent noise accumulator models, where \theta_{y} is fit for every neuron individually, and as such, these parameters are informed by relatively little information which might increase the uncertainty on these parameters. Are the results shown in Figure 4B not potentially an overfitting issue?

In light of the above analysis, we believe the same argument will hold for the independent noise model. Additionally, we do not believe overfitting to be an issue for this model, in the way that the reviewer is concerned about, because both models have an identical number of parameters, a point that we perhaps explained poorly and may have been misunderstood by the reviewer. We have modified the relevant text to make this more clear.

“It is worth emphasizing that the independent noise model is identical to the shared noise model in the way it is parameterized (i.e. number and form of the model parameters) but only differs in the structure of the latent accumulation noise.”

A related question here pertains to the cross-validation procedure: How exactly are the data partitioned? If the parameters fit on the individual neuron level, then the split between train and test data should only split trials of the same neurons (under the independent noise model, \theta_{1,T} cannot be expected to predict \theta_{2,T}, as these are different neurons, I would think? Or am I missing something?).

This is a good question, thank you for bringing it up. We have added the following text to the Methods section to better explain our cross-validation procedure.

“Because most of our models were fit simultaneously to data from multiple experimental sessions (in which different neurons are recorded), to perform cross-validation, we randomly divided trials within each session into a train and test set, and trained and tested the model collectively on those groups of trials. Testing the model in this way will determine parameter robustness across all sessions (for model parameters that are shared across all sessions) and individual parameter robustness within a session (for parameters that are specific to an individual session). This procedure also worked for the ‘independent noise model’, for which model parameters were shared across all sessions, but individual neuron parameters were session specific.”

6. Page 39: were the bounds of optimization ever reached?

The bounds were occasionally reached for specific datasets, and common themes emerged for the different models (e.g., fitting to choice only, or a jointly fit model) about when the bounds were reached. We have augmented the relevant section to make note of this finding:

“The occurrence of parameters hitting the bound can be seen in Figure 3 & Figure 3 — figure supplement 4. The most common boundary hitting situation was a variance parameter (σi, σa, σs) hitting the lower boundary of zero, which means that the model did not support noise of that kind in the model fit. σi and σa were found to do this for the choice only model, consistent with the results of Brunton et al. The other bound that was frequently hit was the upper bound for the accumulation bound parameter *B*, a result also consistent with the results of Brunton et al. The log-likelihood surface as *B* grows very large becomes very flat, because it becomes increasingly unlikely that probability mass *P(a(t))* crosses the boundary. Thus, the model fits do not change appreciably if this optimization boundary is relaxed.”

7. Null joint model: This is related to the point above, but I'm not sure if Figure S9B (referred to on page 42) actually shows the results of this model. I was indeed wondering how well this model cross-validates, in light of the potential collinearity issue raised in point 3 above.

We apologize for the confusion here. Figure S9B (now titled Figure 3 — figure supplement 4) does not explicitly show the results of the null joint model. It compares the model fit improvement of each model over a null model when using subsets of the data (as one would do during multi-fold cross-validation) to the model fit improvement when using all of the data. The purpose of doing this is to justify our choice of illustrating results (e.g. parameters and data synthesize from best fitting models) from models fit to *all* the data. (In other words, any model fit to subsets of the data does not differ appreciably to a model fit to *all* the data.) Figure S9B only shows the results of the null joint model in the sense that it less accurately captures the data compared to the fitted joint model (i.e. the log likelihood is greater than zero for the joint model).

Figure S9B does illustrate that the joint model cross-validates well, i.e., the cross-validated log likelihood per fold when comparing the joint model to the null model is greater than zero, indicating that the joint model is not overparameterized, consistent with our discussion and analysis of this issue in point 5. We believe this provides strong evidence that our null models were appropriately designed — our model justifies the use of the additional parameters our model provides. We have added text relevant to the joint models regarding this point to the Methods:

“The improved performance (i.e. cross-validated log likelihood) of our joint model over the null model shown in Figure 3 — figure supplement 4 further confirms that θna(t) and θ0n(t) are uniquely identifiable, and that they are not redundant (i.e. the joint model is not overparameterized).”

8. Figure S9A: I don't see '+'-symbols.

We have modified the figure so what was formerly a ‘+” is now a diamond, which is easier to see.

9. Typos.Page 21, Figure S1C title: "Example recovered parameters" (missing e in parameters).Page 36: "The transition matrix M(θa, δt) it is determined using methods established in Brunton (2013)". "it" should probably be removed.

Thank you — corrected!

10. Code availability:The authors state they will make the code publicly accessible upon publication. It would be useful to include a persistent link (to e.g. osf or github) in the manuscript to facilitate finding the manuscript after it has been published. The code itself could still remain under embargo as long as the review lasts, should the authors prefer this.

We strongly agree with the reviewers. We have updated our code availability statement to reflect the persistent links to the core codebase for fitting models described, and a second repository for reproducing the results of the manuscript.